



# Elevated oxidized mercury in the free troposphere: Analytical advances and application at a remote continental mountaintop site

Eleanor J. Derry[1*], Tyler Elgiar[2‡], Taylor Y. Wilmot[3], Nicholas W. Hoch[1§], Noah S. Hirshorn[3+], Peter Weiss-Penzias[4], Christopher F. Lee[5], John C. Lin[3], A. Gannet Hallar[6], Rainer Volkamer[5], Seth N. Lyman[2,7], Lynne E. Gratz[1#]

[1]Environmental Studies Program, Colorado College, Colorado Springs, CO 80903 USA
[2]Bingham Research Center, Utah State University, Vernal, UT 84078 USA
[3]Department of Atmospheric Sciences, University of Utah, Salt Lake City, UT 84112 USA
[4]University of California, Santa Cruz, Santa Cruz, CA 95064 USA
[5]Department of Chemistry & CIRES, University of Colorado Boulder, Boulder, CO 80309 USA
[6]Storm Peak Laboratory, Department of Atmospheric Sciences, University of Utah, Salt Lake City, UT 84112 USA
[7]Department of Chemistry and Biochemistry, Utah State University, Logan, Utah, 84322 USA
[*]*Current affiliation:* Department of Chemistry, Reed College, Portland, OR 97202 USA
[‡]*Current affiliation:* Bureau of Land Management, Vernal, UT 84078 USA
[§]*Current affiliation:* BBA Water Consultants, Inc., Englewood, CO 80110 USA
[+]*Current affiliation:* Ramboll, New York, NY 10119 USA
[#]*Current affiliation:* Department of Chemistry and Environmental Studies Program, Reed College, Portland, OR 97202 USA

*Correspondence to:* Lynne E. Gratz (lgratz@reed.edu)

**Abstract:** Mercury (Hg) is a global atmospheric pollutant. In its oxidized form ($Hg^{II}$), atmospheric Hg can readily deposit to ecosystems, where it may bioaccumulate and cause severe health effects. High $Hg^{II}$ concentrations are reported in the free troposphere, but spatiotemporal data coverage is limited. Underestimation of $Hg^{II}$ by commercially available measurement systems hinders quantification of Hg cycling and fate. During spring-summer 2021 and 2022, we measured elemental ($Hg^0$) and oxidized Hg using a calibrated dual-channel system alongside trace gases, aerosol properties, and meteorology at the high-elevation Storm Peak Laboratory (SPL) above Steamboat Springs, Colorado. Oxidized Hg concentrations displayed temporal behavior similar to previous work at SPL, but were approximately three times higher in magnitude due to improved measurement accuracy. We identified 18 multi-day events of elevated $Hg^{II}$ (mean enhancement: 36 pg m$^{-3}$) that occurred in dry air (mean ± s.d. RH = 32 ± 16%). Lagrangian particle dispersion model (HYSPLIT-STILT) 10-day back-trajectories showed that the majority of transport prior to events occurred in the low to mid-free troposphere. Oxidized Hg was anticorrelated with $Hg^0$ during events, with an average (± s.d.) slope of -0.39 ± 0.14, suggestive of upwind oxidation followed by deposition during transport. Concurrent sulfur dioxide measurements verified that three upwind coal-fired power plants did not measurably contribute ambient Hg at SPL. Principal Components Analysis revealed $Hg^{II}$ consistently inversely related with $Hg^0$ and was generally not associated with combustion tracers, confirming oxidation in the clean, dry free troposphere as its primary origin.



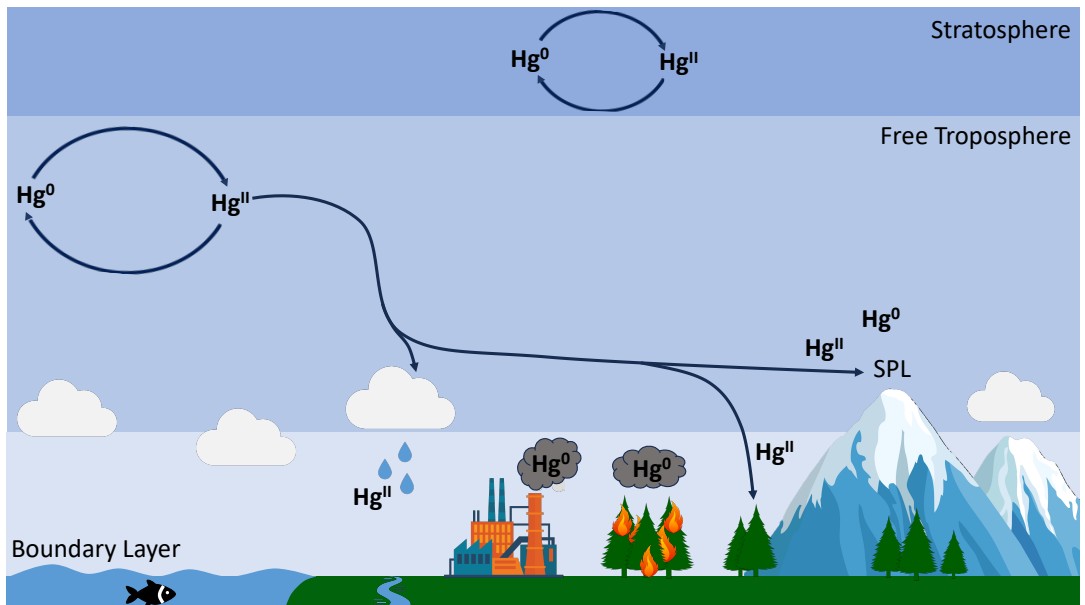

## 1. Introduction

Mercury (Hg) is a global pollutant that can be emitted to the atmosphere from both natural and anthropogenic sources. Humans have changed the Hg biogeochemical cycle through industrial development and land use practices that have increased atmospheric Hg concentrations and altered reservoir distributions (Obrist et al., 2018; Driscoll et al., 2013; Selin, 2009). Mercury is a toxin that can cause neurological and cardiovascular health effects depending on the duration, magnitude, and chemical form of exposure (Lyman et al., 2020a). In the atmosphere, Hg exists as gaseous elemental Hg ($Hg^0$; GEM), gaseous $Hg^{II}$–commonly referred to as GOM (gaseous oxidized mercury) or RGM (reactive gaseous mercury)–and particulate-bound mercury (PBM). Elemental Hg is relatively inert, and has an atmospheric lifetime on the scale of months (Bishop et al., 2020). Oxidized Hg ($Hg^{II} =$ GOM + PBM), however, is much more reactive and water soluble, resulting in an atmospheric lifetime on the scale of days to a week in the planetary boundary layer (PBL) (Lyman et al., 2020a). Thus, when $Hg^0$ undergoes oxidation to form $Hg^{II}$, it is much more readily deposited into ecosystems, where it can methylate and bioaccumulate within food systems, with potential environmental and health consequences (Driscoll et al., 2013; Selin, 2009).

While international and domestic regulations have led to decreases in global background ambient Hg concentrations (Obrist et al., 2018; Lyman et al., 2020a), local conditions can vary significantly due to differences in the magnitude of urban and industrial emissions (Driscoll et al., 2013). Global background terrestrial $Hg^0$ concentrations also vary spatially; concentrations in the northern hemisphere reportedly range from 1.5 to 1.7 ng m$^{-3}$, and from 1.0 to 1.3 ng m$^{-3}$ in the southern hemisphere (Sprovieri et al., 2016) due to a higher concentration of urban areas and greater anthropogenic emissions in the Global North (Mao et al., 2016). Mercury species have also been shown to exhibit variability with altitude. Elemental Hg is typically well mixed in the planetary boundary layer



(PBL), while Hg[II] has been shown to increase in concentration with elevation (Swartzendruber et al., 2006; Faïn et al., 2009; Lyman and Jaffe, 2012; Gratz et al., 2015; Shah et al., 2016). The atmosphere is considered to be a minor reservoir of Hg (~5 GT) compared to soil (1,450 GT) and marine ecosystems (280 GT), but it is the dominant pathway for Hg inputs to ecosystems via deposition (Driscoll et al., 2013; Obrist et al., 2018; Lyman et al., 2020a).


The chemical oxidation–reduction mechanisms of Hg in the atmosphere, which determine its environmental fate, are complex and not fully understood (Dibble et al., 2020; Lyman et al., 2020a; Shah et al., 2021; Castro et al., 2022). Previous studies have indicated multiple possible major oxidants of Hg in the atmosphere. The most recent studies have suggested that Hg oxidation occurs primarily in the free troposphere, and the leading oxidants are halogens such as atomic bromine (Br) and the hydroxyl radical (OH) (Dibble et al., 2020). Oxidation in the free troposphere is thought to be driven by a two-step mechanism, in which ozone acts as a secondary oxidant (Shah et al., 2021; Castro et al., 2022). Previous studies have seen Br-initiated oxidation in the free troposphere (Gratz et al., 2015; Coburn et al., 2016). A companion paper to this study further demonstrated that iodine-initiated oxidation may compete with Br- and OH-initiated oxidation at cold temperatures, and may be important to understanding the Hg oxidation mechanism (Lee et al., in review). This chemical cycling creates a pool of Hg[II] within the free troposphere (Lyman and Jaffe, 2012; Shah et al., 2016; Weiss-Penzias et al., 2015).



Large uncertainties exist in the rate constants of the oxidation mechanism, and there remains a shortage of experimental data (Castro et al., 2022). Part of this uncertainty comes from limitations in commercial instrument and measurement accuracy (Jaffe et al., 2014; Lyman et al., 2020b; Gustin et al., in review). Most measurements of atmospheric Hg[II] to date have relied on KCl-denuders, which exhibit a low bias (Lyman et al., 2020b and references therein). The extent of this low bias cannot be directly quantified, as most Hg[II] measurements have been uncalibrated (Gustin et al., 2015; Jaffe et al., 2014). Thus, these datasets likely suffer from an underestimation of atmospheric Hg[II] concentrations (Lyman et al., 2020b). Another companion paper to this study seeks to correct this underestimation using co-located wet deposition measurements at U.S. AMNet sites (Weiss-Penzias et al., in review).



Previous studies at mountaintop observatories in the U.S., Taiwan, and France have examined temporal trends in atmospheric Hg concentrations and consistently showed evidence of high concentrations of Hg[II] in the clean, dry air of the free troposphere (Swartzendruber et al., 2006; Faïn et al., 2009; Sheu et al., 2010; Timonen et al., 2013; Fu et al., 2016). One such site is Storm Peak Laboratory (SPL), a high-elevation, continental research station in the U.S. Rocky Mountains. Past work at SPL documented transitions between the PBL and the free troposphere using long-term measurements of aerosols and trace gases (Collaud Coen et al., 2018). Moreover, studies by Obrist et al. (2008) and Faïn et al. (2009) investigated the influence of anthropogenic Hg sources, as well as the effects of meteorology and air mass chemical composition on speciated Hg compounds. However, these and other mountaintop studies historically relied on instrumentation that likely underestimated Hg[II] concentrations (Lyman et al., 2020b).



In this study, we employed a calibrated Hg measurement technique with higher time resolution and improved measurement accuracy compared to other available methods (Lyman et al., 2020b; Elgiar et al., in review). Data were collected at SPL above Steamboat Springs, Colorado during two six-month periods in spring and



summer 2021 and 2022. We examined meteorology, air mass composition, and atmospheric transport during periods
of elevated $Hg^{II}$ to more accurately quantify the concentrations of $Hg^{II}$ and to identify its origins in a continental
atmosphere.

**2. Methods**

*2.1 Site Description*

The data used in this study were collected at Storm Peak Laboratory (3220 m AMSL; 40.455 N, 106.744
W) above Steamboat Springs, CO. SPL is a permanent high elevation research facility within the Rocky Mountains
along the Continental Divide. The site is optimally located to characterize the remote continental atmosphere and
transitions between the PBL and the free troposphere (Faïn et al., 2009; Collaud Coen et al., 2018). SPL receives
prevailing westerly winds, creating a clear upwind fetch (Faïn et al., 2009). The site is located east of the agricultural
Yampa Valley and approximately 19 km from downtown Steamboat Springs (Fig. A1). SPL is also located east and
downwind of three coal-fired power plants, located in Hayden and Craig, CO and Vernal, UT, but otherwise sits in a
relatively remote location with few nearby point sources that could influence atmospheric composition at the
laboratory.

*2.2 Data Collection*

*2.2.1 Dual-channel measurements of $Hg^0$ and $Hg^{II}$*

The Utah State University (USU) dual-channel Hg measurement system operated at SPL from March 12,
2021 to October 11, 2021 and March 3, 2022 to September 22, 2022. The operation, validation, and quality
assurance/control of this system at SPL are described in detail in Elgiar et al. (in review). Briefly, the dual-channel
system pulls ambient air through the main Teflon-coated aluminum inlet at a rate of 9 standard L $min^{-1}$ into a
weatherproof box containing a thermal converter and a pair of in-series cation-exchange membranes. The thermal
converter is constructed of quartz, packed with quartz chips, and maintained at a temperature of 650°C to convert
$Hg^{II}$ to $Hg^0$, such that total Hg is measured (THg = $Hg^{II}$ + $Hg^0$) (Lyman et al., 2020b). The cation-exchange
membranes remove $Hg^{II}$ from the sample air stream, allowing only $Hg^0$ to pass through (Miller et al., 2019). A valve
switches between the thermal converter and the cation-exchange membranes every five minutes. During each five-
minute period, two 2.5-minute measurements are recorded by the downstream Tekran 2537X $Hg^0$ vapor analyzer.
Oxidized Hg concentrations are computed as the difference between (a) two consecutive 2.5-minute THg
measurements averaged together, and (b) the average of the 2.5-min $Hg^0$ measurement preceding and the one
following the consecutive THg measurements. As such, the system generates a complete set of Hg measurements
(THg, $Hg^0$, $Hg^{II}$) every ten minutes. Inlet and sample lines are maintained at a temperature of 110°C to minimize
contamination and wall losses.

Elemental mercury vapor injections on the Tekran 2537X were performed every 6 to 8 weeks using a
Tekran 2505 calibration unit to verify the permeation rate of the internal calibration source. The cation exchange



membranes and soda lime trap that was installed upstream of the 2537X to prevent passivation of the internal gold traps, were replaced every two weeks, while the inlet was replaced every four weeks. The dual-channel system was

also verified for measurement accuracy with an SI-traceable calibrator that injects known amounts of $Hg^0$, $HgBr_2$, and $HgCl_2$ into the inlet on a weekly basis, as described in Elgiar et al. (in review). All final $Hg^0$ and $Hg^{II}$ concentrations were increased by 8% to account for a suspected bias in the Dumarey equation used for calculating vapor pressure of $Hg^0$ for manual injections (Elgiar et al., in review; de Krom et al., 2021). One-hour average detection limits for $Hg^{II}$ measurements were $11.8 \pm 6.5$ pg m$^{-3}$ (mean $\pm$ 95% confidence interval of weekly detection

limit tests conducted throughout the measurement season) in 2021 and $5.8 \pm 1.9$ pg m$^{-3}$ in 2022. The detection limits were calculated as three-times the standard deviation of measurements of $Hg^{II}$ during times when both channels were sampling $Hg^{II}$-free air. The percent standard uncertainty for $Hg^0$ and $Hg^{II}$ with the dual-channel system was 8% (Elgiar et al., in review). The placement of the dual-channel component of the system (e.g. the box containing the thermal converter and cation exchange membranes) outdoors and immediately upstream of the inlet, as well as the

development of an automated calibration system for $Hg^0$ and $Hg^{II}$ compounds were both key improvements to the dual-channel system for the present study.

The dual-channel system may occasionally report negative values for $Hg^{II}$ when the $Hg^0$ concentration is greater than the corresponding THg measurement used in the difference calculation (Dunham-Cheatham et al., 2023). This may occur in plumes of rapidly changing concentrations, or for other reasons related to instrument

performance that are as yet not fully understood. In this study, negative concentrations of hourly averaged $Hg^{II}$ were computed only intermittently in between May 2–11 2021, most notably during two approximately 24-hour periods between May 2 and May 4, and for approximately 10 hours on May 10. We therefore excluded several pairs of hourly averaged $Hg^{II}$ and $Hg^0$ concentrations during this early May 2021 period (n = 65; 2% of the March 12 – September 15, 2021 data). Removing these points did not change the 2021 mean, median, or standard deviation of

$Hg^0$ within the measurement precision shown in Table 1; the spring 2021 mean $\pm$ s.d. decreased from $1.32 \pm 0.10$ to $1.31 \pm 9$ ng m$^{-3}$. For $Hg^{II}$, the exclusion of these values increased the 2021 mean $\pm$ s.d. from $101 \pm 52$ (median = 101) pg m$^{-3}$ to $103 \pm 49$ (median = 102) pg m$^{-3}$, and increased the spring 2021 mean $\pm$ s.d. from $77 \pm 54$ (median = 67) pg m$^{-3}$ to $82 \pm 49$ (median = 70) pg m$^{-3}$. In 2022, there was only one negative value for the hourly averaged $Hg^{II}$ concentrations, and that value along with the corresponding $Hg^0$ concentration were also removed. Other extended

gaps in the datasets, related to instrument malfunction or operator error, included the periods from May 12 to June 6, June 29 to July 9, and August 2–10 in 2021, as well as June 27 to July 1 and August 16–23 in 2022.




**Table 1: Summary statistics of Hg$^0$, Hg$^{II}$, trace gases, and aerosol PM$_1$ scattering ($\sigma_{sp}$) measurements by season and for each study year at SPL.**

| | | 2021 | | | 2022 | | |
|---|---|---|---|---|---|---|---|
| | | Spring | Summer | All | Spring | Summer | All |
| Hg$^0$ | mean ± σ | 1.31 ± 0.9 | 1.24 ± 0.14 | 1.27 ± 0.13 | 1.26 ± 0.12 | 1.25 ± 0.10 | 1.25 ± 0.11 |
| (ng m$^{-3}$) | median | 1.34 | 1.25 | 1.30 | 1.29 | 1.25 | 1.27 |
| | max | 1.66 | 2.38 | 2.38 | 1.66 | 1.67 | 1.67 |
| | N | 1332 | 1727 | 3059 | 1718 | 2061 | 3779 |
| Hg$^{II}$ | mean ± σ | 82 ± 49 | 120 ± 41 | 103 ± 49 | 80 ± 40 | 86 ± 29 | 83 ± 35 |
| (pg m$^{-3}$) | median | 70 | 115 | 102 | 73 | 85 | 80 |
| | max | 520 | 253 | 520 | 239 | 197 | 239 |
| | N | 1323 | 1709 | 3032 | 1694 | 2053 | 3747 |
| O$_3$ | mean ± σ | 48 ± 9 | 57 ± 8 | 53 ± 9 | 51 ± 6 | 53 ± 6 | 52 ± 6 |
| (ppb) | median | 50 | 57 | 54 | 51 | 53 | 52 |
| | max | 69 | 88 | 88 | 80 | 84 | 84 |
| | N | 1934 | 2473 | 4407 | 2208 | 2409 | 4617 |
| NO$_x$ | mean ± σ | 1.9 ± 0.5 | 2.1 ± 0.7 | 2.1 ± 0.6 | 1.3 ± 1.0 | 1.3 ± 0.6 | 1.3 ± 0.8 |
| (ppb) | median | 1.9 | 2.0 | 1.9 | 1.0 | 1.2 | 1.1 |
| | max | 12 | 15 | 15 | 18 | 6.3 | 18 |
| | N | 1928 | 2473 | 4401 | 2207 | 2567 | 4774 |
| SO$_2$ | mean ± σ | 0.0 ± 0.1 | 0.0 ± 0.2 | 0.0 ± 0.2 | 0.0 ± 0.3 | 0.0 ± 0.2 | 0.0 ± 0.2 |
| (ppb) | median | 0.0 | 0.0 | 0.0 | 0.0 | 0.0 | 0.0 |
| | max | 1.1 | 3.3 | 3.3 | 5.8 | 3.2 | 5.8 |
| | N | 1934 | 2540 | 4474 | 2195 | 2409 | 4604 |
| CO | mean ± σ | NA | 201 ± 94 | NA | 133 ± 20 | 123 ± 24 | 128 ± 23 |
| (ppb) | median | NA | 175 | NA | 134 | 121 | 128 |
| | max | NA | 1859 | NA | 211 | 276 | 276 |
| | N | NA | 1586 | NA | 2208 | 2568 | 4776 |
| PM$_1$ $\sigma_{sp}$ | mean ± σ | 5 ± 3 | 55 ± 78 | 34 ± 64 | 4 ± 3 | 11 ± 10 | 7 ± 8 |
| (Mm$^{-1}$) | median | 5 | 27 | 9 | 3 | 8 | 5 |
| | max | 17 | 816 | 816 | 33 | 164 | 164 |
| | N | 1849 | 2476 | 4325 | 2175 | 2460 | 4635 |

*2.2.2 Criteria Gas and Meteorological Measurements*

Several criteria gases and meteorological parameters were continuously measured at SPL related to this study. Meteorological data were measured on the roof of SPL at a height of 10 m above ground level (AGL) at a five-minute time resolution. Measured meteorological parameters included temperature, relative humidity (RH), wind speed and direction, and barometric pressure. These data were quality-assured and made publicly available by
MesoWest (https://mesowest.utah.edu). We computed water vapor mixing ratio using a combination of measured temperature, RH, and barometric pressure, and the theoretical expression of the Clausius-Clapeyron equation.

Ozone (O$_3$), nitrogen oxides (NO$_x$), sulfur dioxide (SO$_2$), and aerosol properties such as scattering and absorption at 450, 550, and 700 nm were measured at a time resolution of one minute, and calibrated daily. Ozone was measured using a Thermo Model 49i Analyzer with a precision of 1.0 ppbv. Nitrogen oxides were measured
with a Thermo Model 42i NO-NO$_2$-NO$_x$ Analyzer with a precision of 0.2 ppbv. Sulfur dioxide was measured using a Thermo Model 43i Analyzer with a precision of the greater value of either 1% or 1 ppbv. Aerosol properties were





measured with a TSI model 3562 Nephelometer. Analysis for this manuscript relied on $PM_1$ scattering ($PM_1$ $\sigma_{sp}$) as the representative aerosol metric, in part for direct comparison with related studies (e.g. Timonen et al., 2013), and because the aerosol absorption data had more frequent gaps. Aerosol data were corrected to STP conditions and

quality controlled/assured by NOAA ESRL GML (Andrews et al., 2019). Carbon monoxide (CO) was measured beginning on July 7, 2021 with a Teledyne model 300E, from which average values were logged every 2.5 minutes in 2021 and every minute in 2022. The analyzer performed an automatic zero every 4 h in CO-free air; routine on-site spans for CO could not be performed due to COVID-19-related site access limitations, but based on those able to be performed before, during, and after the study, the measurement accuracy was estimated to be within $\pm$ 25%.

*2.3 Data Analysis & Modeling Techniques*

*2.3.1 Statistical Treatment of Data*

The measurement data were averaged to one-hour intervals corresponding to the beginning of each hour to compare all the criteria on the same time step. Analyses in the present manuscript focus specifically on measurements made from March 13, 2021 to September 15, 2021 and from March 3, 2022 to September 15, 2022 to

limit the analysis to two seasons (spring and summer), while also encompassing a full six months of data in 2021. We defined spring as 1-March to 31-May and summer as 1-June to 15-September given prior knowledge of the seasonal climatology and transport patterns (Obrist et al., 2008; Hallar et al., 2016). For example, past work has shown that trans-Pacific transport as well as stratospheric subsidence occur more commonly in springtime (Hallar et al., 2016), while summertime air masses at SPL are frequently impacted by biomass burning and different transport

patterns (Obrist et al., 2018). Additionally, Hg concentrations have been shown to vary seasonally within the northern hemisphere, driven largely by seasonal meteorology (Xu et al., 2022; Custódio et al., 2022).

In this study, statistical significance was defined as $p < 0.05$. Reduced major axis (RMA) regressions were used to calculate linear regression slopes to account for uncertainty in both variables, as recommended for air quality data (Ayers, 2001). Correlation analysis was performed using Pearson's correlation coefficients (R), and

comparisons of means were calculated using two-tailed independent sample t-tests and Mann-Whitney U tests.

Lastly, we estimated that at least one third of the ten-minute-averaged measurements in June – September 2021 showed evidence of smoke presence, whereas this was detected in less than 5% of data for the same timeframe in 2022. The presence of smoke was based on preliminary criteria of CO $\geq$ 150 ppbv, $PM_1$ $\sigma_{sp} \geq$ 30 $Mm^{-1}$ and $PM_{10}$ $\sigma_{sp} \geq$ 35 $Mm^{-1}$ for a minimum of one hour, and confirmation of overhead smoke using the National Oceanic and

Atmospheric Administration Hazard Mapping System Fire and Smoke Product (NOAA HMS) (Air-Now Tech, 2023).

*2.3.2 Identification of Events of Elevated Oxidized Mercury*

Events of elevated $Hg^{II}$ were initially defined as time periods when $Hg^{II}$ concentrations exceeded the seasonal mean by at least one standard deviation (Table 1) for a minimum of 24 hours. Adjacent periods of elevated

$Hg^{II}$ were counted as the same event if they appeared to represent the same air mass, based on concurrent





meteorological and trace gas measurements (e.g., consistently similar Hg⁰, Hgᴵᴵ, and trace gas concentrations and
RH before and after a brief intrusion of air associated with the boundary layer). Additional hours were then included
in the events at the beginning and end of periods of high Hgᴵᴵ in order to capture the transition between air mass
conditions. In total, we identified 18 events of prolonged high Hgᴵᴵ in the 2021 and 2022 measurement periods

(Table 2). All events had at least 85% Hg data coverage. We characterized the events through statistical analysis of
Hg, trace gases, and meteorology as well as air mass transport analysis (Sect. 3.2.1 and 3.2.2) in order to understand
the origins of the air masses containing elevated concentrations of Hgᴵᴵ. We also analyzed the June 2022
measurement period as a case study because it contained relatively continuous records of all measured species and
included five distinct events of high Hgᴵᴵ (Events 11–15; Table 2), separated by periods of depleted Hgᴵᴵ that were

defined as non-events 1–5 (Sect 3.2.3). Non-Event 5 was made to include the 24 hours following the end of Event
15. Analysis of the measurement data within and between events was complemented by a simulation of air mass
origins, as described below.






**Table 2: Event mean ± s.d. for Hg$^{II}$, Hg$^0$, RH, O$_3$, CO, and PM$_1$ σ$_{sp}$. [†]These events had evidence of smoke from local or regional biomass burning.**

| Event | Date (MST) | Hg$^0$ (ng m$^{-3}$) | Hg$^{II}$ (pg m$^{-3}$) | RH (%) | O$_3$ (ppb) | CO (ppb) | PM$_1$ σ$_{sp}$ (Mm$^{-1}$) |
|---|---|---|---|---|---|---|---|
| 1 | 4/1/21 12:00 – 4/6/21 06:00 | 1.12 ± 0.07 | 137 ± 38 | 26 ± 8 | 40 ± 3 | NA | 4.5 ± 0.6 |
| 2 | 4/24/21 23:00 – 4/27/21 05:00 | 1.18 ± 0.06 | 121 ± 25 | 38 ± 16 | 53 ± 3 | NA | 5.0 ± 1.5 |
| 3 | 4/30/21 02:00 – 5/2/21 00:00 | 1.22 ± 0.06 | 115 ± 47 | 34 ± 7 | 51 ± 3 | NA | 2.6 ± 0.3 |
| 4 | 6/7/21 14:00 – 6/11/21 08:00[†] | 1.03 ± 0.15 | 174 ± 52 | 20 ± 7 | NA | NA | 18.0 ± 21.4 |
| 5 | 6/13/21 23:00 – 6/15/21 21:00[†] | 1.03 ± 0.09 | 156 ± 46 | 13 ± 4 | 49 ± 4 | NA | 20.3 ± 18.5 |
| 6 | 8/22/21 12:00 – 8/30/21 23:00[†] | 1.08 ± 0.08 | 139 ± 31 | 36 ± 16 | 52 ± 5 | 150 ± 30 | 19.3 ± 21.6 |
| 7 | 9/6/21 05:00 – 9/11/21 05:00[†] | 1.12 ± 0.07 | 173 ± 28 | 20 ± 7 | 63 ± 5 | 230 ± 50 | 49.5 ± 21.6 |
| 8 | 9/13/21 19:00 – 9/16/21 07:00[†] | 1.29 ± 0.10 | 163 ± 32 | 35 ± 14 | 63 ± 6 | 160 ± 20 | 23.1 ± 9.7 |
| 9 | 5/10/22 00:00 – 5/12/22 15:00 | 1.09 ± 0.10 | 135 ± 50 | 35 ± 15 | 54 ± 5 | 120 ± 10 | 2.7 ± 0.8 |
| 10 | 5/25/22 19:00 – 5/27/22 18:00 | 1.13 ± 0.08 | 120 ± 34 | 39 ± 10 | 49 ± 3 | 110 ± 20 | 5.2 ± 1.2 |
| 11 | 6/3/22 04:00 – 6/5/22 04:00 | 1.08 ± 0.08 | 100 ± 21 | 38 ± 10 | 56 ± 3 | 110 ± 20 | 3.9 ± 1.1 |
| 12 | 6/7/22 00:00 – 6/11/22 17:00 | 1.18 ± 0.07 | 124 ± 19 | 38 ± 10 | 57 ± 7 | 130 ± 10 | 4.9 ± 1.4 |
| 13 | 6/12/22 18:00 – 6/13/22 23:00[†] | 1.09 ± 0.10 | 117 ± 46 | 38 ± 30 | 55 ± 5 | 110 ± 40 | 21.3 ± 31.0 |
| 14 | 6/15/22 17:00 – 6/17/22 15:00[†] | 1.12 ± 0.07 | 118 ± 13 | 28 ± 11 | 59 ± 7 | 130 ± 30 | 13.7 ± 10.0 |
| 15 | 6/20/22 07:00 – 6/23/22 02:00 | 1.12 ± 0.04 | 116 ± 20 | 30 ± 9 | 60 ± 5 | 120 ± 10 | 7.0 ± 3.6 |
| 16 | 7/20/22 07:00 – 7/22/22 23:00 | 1.21 ± 0.08 | 120 ± 12 | 35 ± 9 | 59 ± 3 | 120 ± 20 | 10.6 ± 2.2 |
| 17 | 7/25/22 00:00 – 7/27/22 23:00 | 1.21 ± 0.06 | 100 ± 30 | 49 ± 21 | NA | 140 ± 30 | 16.8 ± 5.8 |
| 18 | 9/11/22 18:00 – 9/13/22 23:00[†] | 1.14 ± 0.05 | 107 ± 22 | 31 ± 25 | 55 ± 6 | 140 ± 20 | 26.9 ± 10.9 |



### 2.3.3 Air Mass Transport Analysis using HYSPLIT-STILT

The Hybrid Single-Particle Lagrangian Integrated Trajectory model integrating features from the Stochastic Time-Inverted Lagrangian Transport model (HYSPLIT-STILT) is a Lagrangian particle dispersion model (Loughner

et al., 2021; Lin et al., 2003) that was used to investigate the history of air masses arriving at SPL. For each of the 18 events and 5 non-events presented here, an ensemble of 1,000 air parcels were released at SPL at three-hour intervals and traced for 240 hours backward in time. The air parcels were initialized at 5 m above ground level and transported with stochastic motions (simulating turbulence) with HYSPLIT-STILT, driven by meteorological fields from the 3 km meteorology from the High-Resolution Rapid Refresh model (HRRR; Dowell et al., 2022; NOAA

ARL, 2024), as well as the 0.25°×0.25° meteorology from the Global Forecast System (GFS; NOAA ARL, 2024). Nesting HRRR meteorology within the relatively coarse GFS meteorological fields was necessary for simulating atmospheric transport over the Pacific Ocean. Along each air parcel's backward trajectory, position, the mixed layer depth, precipitation rate, RH, temperature, and total cloud cover from the meteorological fields were sampled at one-minute intervals, providing insight into the spatial origin and meteorological conditions associated with each air

mass.

### 2.3.4 Principal Components Analysis

We used the Principal Components multivariate factor analysis technique (PCA) to investigate the interrelationships between measured variables and identify broad patterns in air mass composition at SPL. Factor analysis methods have been utilized in several studies involving continuous atmospheric measurements in both

urban/industrial and remote environments (Swartzendruber et al., 2006; Liu et al., 2007; Lynam and Keeler, 2006; Tokarek et al., 2018). Other methods such as positive matrix factorization (PMF) or non-negative matrix factorization (NMF) are recommended for quantitative identification of source-receptor relationships (Hopke and Jaffe, 2020) and are often applied in urban/industrial environments using data from a common instrument (e.g. a suite of VOC measurements, as in Peng et al., 2022 and Gkatzelis et al., 2021). Though PCA is an unweighted least-

squares method (Hopke and Jaffe, 2020), it has the advantage in the present study as an exploratory approach that can indicate both the magnitude and sign of the statistical relationships between variables (Jolliffe and Cadima, 2016). PCA is also advantageous at SPL where the combined dataset consisted of continuous parameters produced by multiple instruments with different measurement scales, and the objective was to use the underlying nature of statistical covariances to broadly characterize air mass compositions at SPL. PCA results were considered in tandem

with the case study analysis to extrapolate the conditions under which enhancements in $Hg^{II}$ or other variables were observed.

We applied PCA with Varimax rotation and Kaiser normalization (IBM SPSS v29.0.1.1) to each of the four sampled seasons in this study. Spring and summer were modeled separately, considering prior knowledge of seasonal climatology and transport (Obrist et al., 2008; Hallar et al., 2016), because air mass composition at SPL in

summer 2021 was intermittently impacted by local or regional wildfire smoke (Sect. 2.3.1), and because the CO instrument was not operational in spring 2021. We additionally modeled the full 2022 measurement period (March 1





– September 15, 2022) because there was notably less evidence for regional wildfire smoke in the air at SPL during summer 2022.

Input variables for PCA included $Hg^0$, $Hg^{II}$, CO, $O_3$, $NO_x$, $PM_1$ $\sigma_{sp}$, water vapor mixing ratio, and barometric pressure. Wind speed was excluded given its weak communalities in the output for most seasons. Although $Hg^{II}$ was generally better correlated with RH (Sect. 3.2.1), water vapor mixing ratio was used in place of RH in PCA as a potentially better indicator of atmospheric moisture content, and because the upper bound of 100% in RH measurements influences its statistical distribution. Moreover, after filtering the data for RH < 85% to exclude periods when SPL may have been in cloud, the Pearson's R and p-values for $Hg^{II}$ vs. water vapor mixing in each season did not substantially change in comparison to $Hg^{II}$ vs. RH, indicating that water vapor mixing ratio was a robust predictor of $Hg^{II}$ even if not as strong as RH (Table A1). Sulfur dioxide was excluded because it was near 0 ppbv on average, with enhancements observed only in short-lived fresh combustion plumes.

Prior to running PCA, variables with |skewness| larger than 0.5 (moderate skew) or larger than 1.0 (high skew) were log-transformed. If the |skewness| did not improve then the non-log-transformed variable was retained. All variables with |skewness| > 1.0 improved following log transform. In most seasons, $PM_1$ $\sigma_{sp}$ also needed to be shifted by its minimum before transformation due to small negative values. Data points three standard deviations above or below the dataset mean were then removed as outliers (< 2% of values). Lastly, variables were standardized (mean = 0, standard deviation = 1) by subtracting that variable's mean and dividing by the standard deviation for the sample period. Data were excluded listwise, which reduced the input datasets by 40% in spring 2021 (n = 1169), 55% in summer 2021 (n = 1147), 29% in spring 2022 (n = 1575), 33% in summer 2022 (n = 1729), and 31% in spring–summer 2022 (n = 3299). The larger percent reductions during 2021 reflect the delayed start of CO measurements and several prolonged maintenance periods; nevertheless, there were ample cases per variable to confirm data suitability for PCA.

Suitable solutions were identified using the Kaiser–Meyer–Olkin (KMO) measure of sampling adequacy for the overall dataset and for individual variables (preferring outputs with KMO > 0.5), and Bartlett's test of sphericity (p < 0.05). Individual variables were also considered for inclusion in the final solution based on extraction communalities > 0.5. The final number of factors was chosen based on the eigenvalue > 1 criteria.

## 3. Results and Discussion

### 3.1 Data Overview

#### 3.1.1 Mercury Overview

Figure 1 and Table 1 summarize the hourly-averaged measurements of Hg and trace gases from the 2021 and 2022 periods. Overall, mean $Hg^0$ concentrations varied minimally from 2021 (1.27 ± 0.13 ng m$^{-3}$) to 2022 (1.25 ± 0.11 ng m$^{-3}$), from spring to summer in each year, or from one season to that same season in the following year, even though t-tests for comparisons of means all indicated statistically significant differences (p < 0.01). The largest mean difference in $Hg^0$ was from spring to summer 2021, but was still less than a 0.1 ng m$^{-3}$ change. Mean $Hg^{II}$





concentrations were not significantly different between the two spring seasons (p = 0.33), whereas the summer 2021 mean was significantly higher (p << 0.001) than in summer 2022.

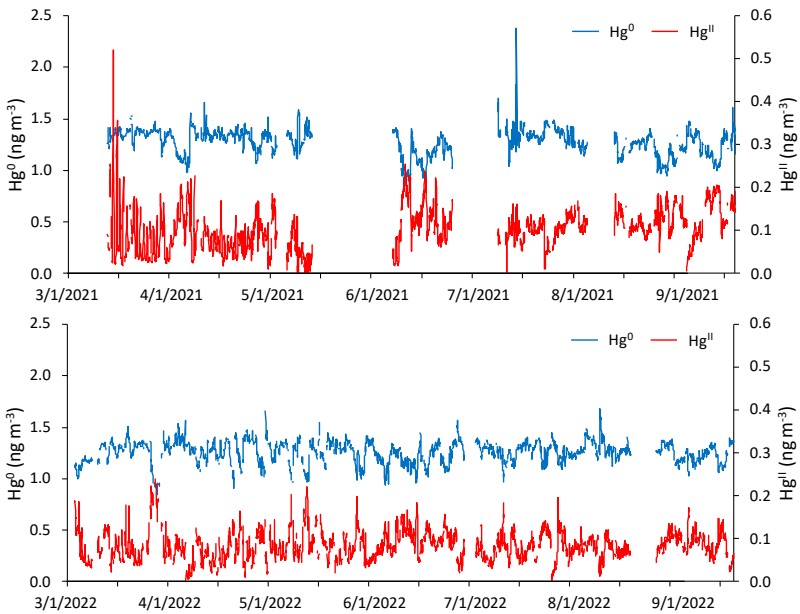

**Figure 1: Time series of hourly-averaged concentrations of Hg$^0$ (blue) and Hg$^{II}$ (red) in units of ng m$^{-3}$**

**between March 1 and September 15 of 2021 and 2022, as measured by the dual-channel system at Storm Peak Laboratory above Steamboat Springs, CO.**

*3.1.2 Trace Gas Overview*

Mean values of O$_3$, NO$_x$, CO, and aerosol scattering were also significantly different (p < 0.01) between the two spring seasons (excluding CO, which was not available in spring 2021) and between the two summer seasons.

Mean values were also significantly different between each spring season and the corresponding summer from that year with the exception of NO$_x$ in 2022 (p = 0.52). It was notable that ozone displayed a relatively flat diel pattern, with mean diel amplitudes of only 4 ppbv in both spring seasons and 2-3 ppbv in the summer seasons, lacking the pronounced daytime enhancement that is commonly seen at lower elevation sites. This behavior was similar to what has been reported at other high elevation sites that were less influenced by daytime production from local precursor

emissions or by nighttime loss processes in the presence of NO$_x$ sources (Mueller, 1994; Monks et al., 2000; Bien and Helmig, 2018; Brodin et al., 2010), suggesting that SPL was routinely influenced by the background free troposphere even in the summertime. Higher mean concentrations of O$_3$, NO$_x$, CO, and aerosol scattering in summer 2021 compared to 2022 may be related, at least in part, to observed differences in the frequency of wildfire smoke presence (Sect. 2.3.1).

We also considered the potential influence of the three upwind coal-fired power plants, located 40, 80, and 200 km west of SPL, on Hg concentrations measured at SPL. These plants were shown to influence air mass



composition at SPL through emissions of $SO_2$ that can contribute to new particle formation (Hallar et al., 2016). A goal was to determine whether elevated $Hg^0$ or $Hg^{II}$ concentrations occurred primarily in the background atmosphere or also under the influence of local or regional point-source emissions. Air masses at SPL were defined as power plant-impacted using the 95th percentile of 10-minute averaged $SO_2$ concentrations from the 2021 measurement period, together with directional analysis using local wind speed and direction measurements at SPL. We also considered several case studies of the highest $SO_2$ concentrations in each season. Smoke-impacted periods (Sect. 2.3.1) were excluded from this analysis. Together, these observations showed that higher concentrations of $SO_2$ reliably corresponded to a source region representative of the three upwind power plants.

Mean $Hg^0$ and $Hg^{II}$ concentrations in spring were statistically significantly different between power plant-impacted and non-impacted air masses; however, the differences were small enough that we did not consider them to be detectable within instrument precision ($\Delta$ mean $Hg^0$ = -0.02 ng m$^{-3}$, $\Delta$ mean $Hg^{II}$ = +4 pg m$^{-3}$). Differences in summertime measurements of $Hg^0$ and $Hg^{II}$ between the two types of air masses were also statistically significantly different but only slightly elevated in power plant-impacted air masses ($\Delta$ mean: $Hg^0$ = +0.04 ng m$^{-3}$, $Hg^{II}$ = +12 pg m$^{-3}$). Yet, the coefficients of determination between $Hg^0$ and $SO_2$ as well as $Hg^{II}$ and $SO_2$ within power plant-impacted periods were very low in both spring and summer ($R^2$ = 0.00 – 0.03) indicating that little to no variation in Hg species could be explained by the variability in $SO_2$. We therefore concluded that the three coal-fired power plants upwind of SPL did not significantly contribute to ambient Hg measurements made at SPL. The lack of a measurable enhancement in Hg at the lab when the coal-fired power plant signature was evident can likely be attributed to power plant emissions controls and lower Hg content coal (Benson, 2003).

### 3.1.3 Comparison to Similar Studies

Mean $Hg^0$ concentrations in this study ($1.27 \pm 0.13$ ng m$^{-3}$ in 2021 and $1.25 \pm 0.11$ ng m$^{-3}$ in 2022) were lower than those reported at other remote mountaintop observatories between 2005–2016, including the Mt. Bachelor Observatory (MBO) in central Oregon, USA ($1.54 \pm 0.176$ ng m$^{-3}$ from May – August 2005; Swartzendruber et al., 2006), the Lulin Atmospheric Background Station (1.73 ng m$^{-3}$ from April 2006 to December 2007; Sheu et al., 2010), the Pic du Midi Observatory in southern France ($1.86 \pm 0.27$ ng m$^{-3}$ from November 2011 – November 2012; Fu et al., 2016), and previously at SPL ($1.51 \pm 0.11$ ng m$^{-3}$ from October 2006 – May 2007 (Obrist et al., 2008); $1.6 \pm 0.3$ ng m$^{-3}$ from April – July 2008 (Faïn et al., 2009)). The observed $Hg^0$ means at SPL in 2021 and 2022 were also lower than estimated global background concentrations in the Northern Hemisphere (1.5–1.7 ng m$^{-3}$; Sprovieri et al., 2016; Mao et al., 2016). One plausible explanation for these differences is the reported declines in ambient $Hg^0$ concentrations in the northern hemisphere from the 1990s to 2005–2013, albeit the magnitudes of reported trends are variable from region to region, ranging from less than 1% to as much as 3.3% per year in northern latitude sites (Lyman et al., 2020a and references therein). Meanwhile Slemr et al. (2011) estimated decreasing trends from 1996 to 2009 of 1.4% and 2.7% per year for the northern and southern hemispheres, respectively. Since the mid-2000s, some studies have reported more modest decreases or even increases in some locations, attributed to variable anthropogenic emission trends and biomass burning, changes in Hg cycling and exchange rates, and the effects of temperature on deposition rates (Lyman et al., 2020a and references therein).



Keeping these spatiotemporal trends in mind, we compared the mean $Hg^0$ concentrations at SPL from Obrist et al. (2008) and Faïn et al. (2009) (~1.56 ng m$^{-3}$) with the mean $Hg^0$ at SPL in 2021 and 2022 (~1.26 ng m$^{-3}$).

Considering this drop of ~0.3 ng m$^{-3}$ over 14 years, it can be estimated that the more recent measurements were lower by ~1.4% per year, a value that is consistent with the range of reported downward trends in northern hemisphere background concentrations. Such a difference could also be related to changes in measurement technology between earlier studies and the present one, i.e. gaseous $Hg^{II}$ (aka, RGM or GOM) that was not retained on the KCl denuder in the earlier SPL studies may have instead been captured downstream as $Hg^0$ resulting in an

overestimate of $Hg^0$ concentrations (Lyman et al., 2010). Even so, total Hg would still have been conserved; yet, the mean (± 1 s.d.) concentrations of THg were only 1.37 ± 0.11 and 1.34 ± 0.09 ng m$^{-3}$ in in 2021 and 2022, respectively, suggesting that SPL may in fact have experienced declining ambient Hg concentrations over time.

In contrast, concentrations of $Hg^{II}$ measured with the dual-channel measurement system in 2021 (103 ± 49 pg m$^{-3}$) and 2022 (83 ± 35 pg m$^{-3}$) (Table 1) were considerably higher than the previous measurements made at SPL

using the KCl-denuder system (Faïn et al., 2009). Summing the April – July 2008 mean measurements of GOM (20 pg m$^{-3}$) and PBM (9 pg m$^{-3}$) in Faïn et al. (2009), it can be estimated that the mean $Hg^{II}$ during that study period was 29 pg m$^{-3}$, which is lower than the spring-summer means in 2021 and 2022 by 3.6- and 2.8-times, respectively. The maximum GOM+PBM concentration Faïn et al. (2009) recorded during GOM enhancement events was 159 pg m$^{-3}$ (Event 4), which is 3.3 times lower than the 2021 maximum (520 pg m$^{-3}$) and 1.5 times lower than the 2022

maximum (239 pg m$^{-3}$). Relatedly, mean measurements of GOM and PBM at other mountaintop sites using the KCl denuder system were respectively 43 pg m$^{-3}$ and 5.2 pg m$^{-3}$ (MBO, USA; Swartzendruber et al., 2006), 12.1 pg m$^{-3}$ and 2.3 pg m$^{-3}$ (LABS, Taiwan; Sheu et al., 2010), and 27 pg m$^{-3}$ and 14 pg m$^{-3}$ (PDM, France; Fu et al., 2016). While these sites occasionally saw large spikes on the order of hundreds of pg m$^{-3}$, it is particularly striking that mean $Hg^{II}$ concentrations in the present study were consistently higher than these other sites by a factor of roughly 2

to 5. These observations have important implications for the accurate representation of total Hg speciation in ambient air. For example, estimates of the fraction of atmospheric THg composed of gaseous $Hg^{II}$ have ranged from 2–20% depending on sampling location and instrumentation (Dunham-Cheatham et al., 2023; Gustin et al., 2023; Osterwalder et al., 2021; Steffen et al., 2008) During events of elevated $Hg^{II}$ in the present study (Sect 3.2), the maximum percent $Hg^{II}$ of THg ranged from 12–22%, with a mean (± s.d.) of 11 ± 2%. Comparatively, the previous

study at SPL using KCl denuders had a mean $Hg^{II}$:THg ratio during enhanced RGM events of approximately 3 ± 1% (Faïn et al., 2009). The recent measurements at SPL using the dual-channel system thus represent a significant contribution to the research field in the ability to report verified, accurate, and reliably higher $Hg^{II}$ concentration measurements than in past studies.

In both years, and for all sampled seasons, $Hg^0$ did not display a pronounced diel pattern (Fig. 2).

Concentrations generally displayed higher concentrations throughout the day in spring 2021 compared to the same hours in summer 2021 and spring 2022, whereas the spread of data at each hour was larger in summer 2021 and spring 2022. Nevertheless, the diel curves were relatively flat across the hours within all sampled seasons. In contrast, $Hg^{II}$ displayed markedly higher concentrations during the spring daytime (Fig. 2), with maxima centered around 11:00–12:00 MST and mean diel amplitudes of 58 pg m$^{-3}$ in spring 2021 and 32 pg m$^{-3}$ in spring 2022. On a





monthly basis (not shown), daytime enhancements were the most pronounced in March 2021 with a mean diel
      amplitude of 117 pg m$^{-3}$. In both years, this pattern diminished throughout the spring and into the summer of 2021,
      with mean diel summer amplitudes of 16 pg m$^{-3}$ (2021) and 19 pg m$^{-3}$ (2022). The diel curves for Hg$^{II}$ also
      emphasize the overall higher concentrations measured throughout all hours of the day in summer 2021, particularly
      overnight, as compared to summer 2022 (Fig. 2, Table 1).

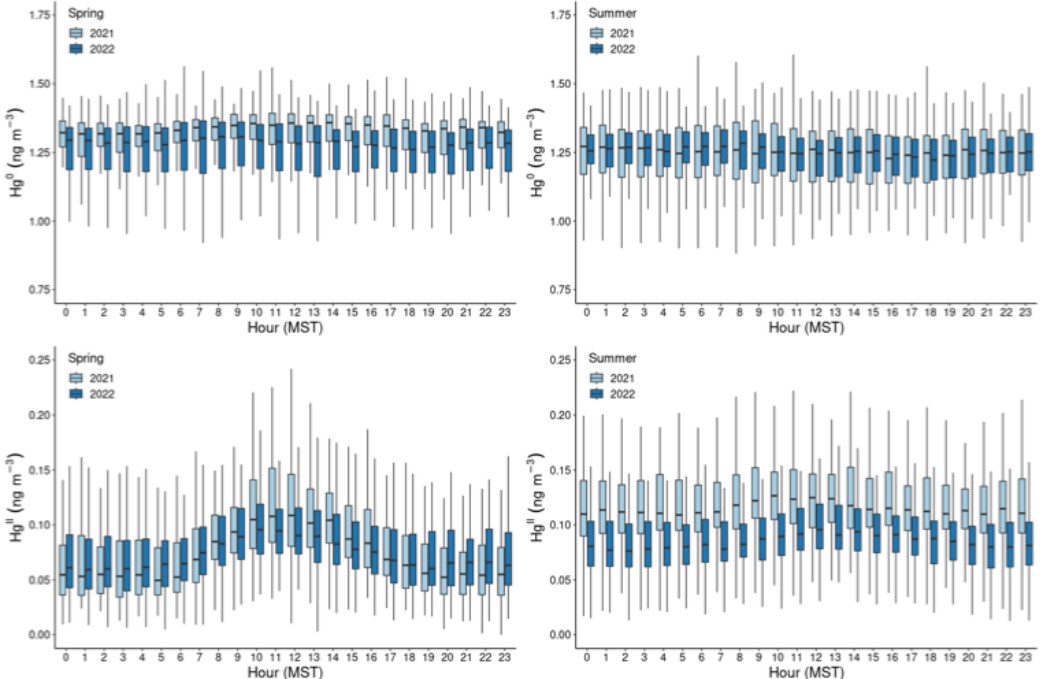


**Figure 2: Boxplots of concentration measurements by hour of day (MST) for Hg$^0$ (top) and Hg$^{II}$ (bottom),
showing diel variability during the spring (left) and summer (right) seasons of the study years. The centerline
of each boxplot represents the median concentration, the box represents the interquartile range, and
upper/lower whiskers are either the maximum/minimum value or the upper/lower quartile value plus/minus**

**1.5 times the interquartile range. Outliers are not shown.**

      Higher daytime concentrations of Hg$^{II}$ were also observed at SPL by Faïn et al. (2009). The authors
attributed this behavior to surface heating processes and uplift of boundary layer air; however, the daytime maxima
during that study tended to occur later in the day, closer to 15:00 MST. Nevertheless, the lack of diel variability in
Hg$^0$ and a daytime peak in Hg$^{II}$ seen in present and past work at SPL is in contrast to other mountaintop studies

which reported higher Hg$^{II}$ overnight and in the early morning hours, typically attributed a shallow planetary
boundary layer and subsidence of Hg$^{II}$-rich air from the upper troposphere and lower stratosphere (UT/LS)
(Swartzendruber et al., 2006; Sheu et al., 2010; Fu et al., 2016). Nighttime subsidence of Hg$^{II}$ from the UT/LS at
MBO, PDM, and LABS was also supported by observations of lower RH, higher ozone, and air mass back-
trajectories pointing to upper tropospheric transport. Further analysis at MBO also identified episodes of high Hg$^{II}$



with transport from the marine boundary layer or trans-Pacific transport of combustion emissions from East Asia
(Timonen et al., 2013). The more continental nature of SPL is one likely factor that contributed to the different
features and transport pathways associated with high $Hg^{II}$ than these other mountaintop sites, as any air masses
influenced by the marine boundary layer or trans-Pacific sources may have been diluted by regional continental air
masses before reaching the site. Faïn et al. (2009) also showed, and we confirmed in the present work (Sect. 3.2),

that high $Hg^{II}$ events at SPL were not associated with UT/LS subsidence and were instead influenced by transport
from within the low to mid-free troposphere, at least in the 10 days of simulated transport history.

*3.2 Multi-day events of enhanced $Hg^{II}$*

*3.2.1 Behaviors in $Hg^0$, $Hg^{II}$, and atmospheric transport*

        Based on the criteria described in Sect. 2.3.2, we identified 18 events of enhanced $Hg^{II}$ during the 2021 and

2022 measurement periods. Three events occurred during spring 2021, five during summer 2021, two during spring
2022, and eight during summer 2022 (Table 2). Across all events, elevated $Hg^{II}$ concentrations were associated with
concurrent decreases in $Hg^0$ and RH (Fig. 3). Relative humidity was consistently low during the event periods, with
a mean (± s.d.) event RH of 32 ± 16%. This observation of high $Hg^{II}$ in very dry air masses is consistent with the
results of Faïn et al. (2009), who concluded that $Hg^{II}$ enhancements at SPL were associated with air masses in the

dry (RH < 40%) free troposphere. In all but Event 10, $Hg^{II}$ was significantly anticorrelated with $Hg^0$ (Pearson's R =
-0.95 to -0.41, p < 0.001), and in all but Event 14, $Hg^{II}$ was significantly anticorrelated with RH (Pearson's R = -
0.81 to -0.28, p < 0.05). Similar to Faïn et al. (2009), $Hg^{II}$ was also anticorrelated with water vapor mixing ratio
during 12 events (Pearson's R = -0.74 to -0.12, p < 0.05). RMA regression slopes (± standard error) for $Hg^{II}$ versus
$Hg^0$ during each event ranged from -0.76 ± 0.01 to -0.16 ± 0.01 (Table 3), with an average event slope (± s.d.) of -

0.39 ± 0.14. Though previous work at SPL also showed negative RGM versus GEM slopes during high RGM
events, the magnitudes were only between -0.07 to -0.18, with an average of -0.10 (Faïn et al., 2009). This
difference likely reflects, at least in part, the improved accuracy of the dual-channel system in measuring $Hg^{II}$ than
previously used instrumentation (Sect. 3.1). The slope of all hourly data during event periods (± standard error) in
2021 and 2022 was -0.38 ± 0.09 (Fig. 3).





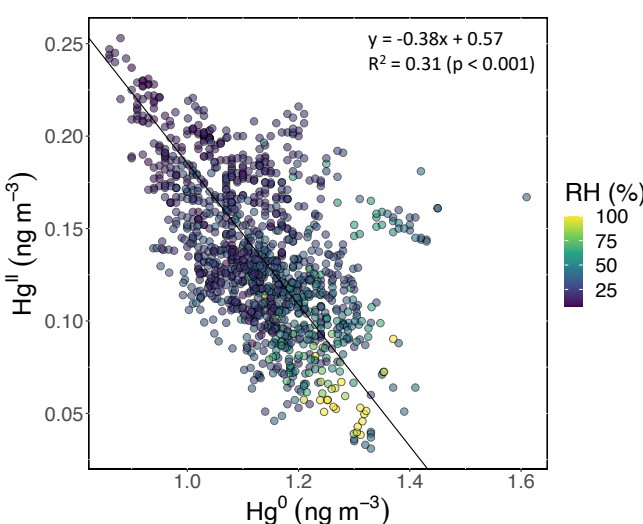


**Figure 3: Hg$^{II}$ vs Hg$^0$ for hourly measurements during all 18 events, color coded by RH. Hg$^{II}$ was significantly anticorrelated with Hg$^0$ (Pearson's R = -0.55, p << 0.001) and RH (Pearson's R = -0.56, p << 0.001). The slope of the RMA regression (± S.E.) was -0.38 ± 0.09, which is similar to the average of all 18 event slopes (± s.d.), which was -0.38 ± 0.14.**









**Table 3: Event RMA regression slopes ± S.E. for $Hg^{II}$ vs. $Hg^0$, Pearson correlation coefficients for $Hg^{II}$ vs. $Hg^0$, RH, $O_3$, and $PM_1 \sigma_{sp}$, and ratios of $Hg^{II}:Hg^0$ and $Hg^{II}:THg$. *p < 0.05, **p < 0.001.**

| Event | m ± S.E. | Pearson's R-value | | | | Hg ratio | |
|---|---|---|---|---|---|---|---|
| | $Hg^{II}/Hg^0$ | $Hg^{II}$ vs $Hg^0$ | $Hg^{II}$ vs RH | $Hg^{II}$ vs $O_3$ | $Hg^{II}$ vs $PM_1 \sigma_{sp}$ | $Hg^{II}:THg$ (%) | $Hg^{II}:Hg^0$ (unitless) |
| 1 | -0.52 ± 0.03 | -0.63** | -0.37** | 0.03 | -0.55** | 11 ± 3 | 0.12 ± 0.04 |
| 2 | -0.45 ± 0.02 | -0.55** | -0.28* | 0.10 | -0.11 | 9 ± 2 | 0.10 ± 0.03 |
| 3 | -0.76 ± 0.01 | -0.95** | -0.53** | 0.71** | 0.59** | 9 ± 4 | 0.10 ± 0.04 |
| 4 | -0.35 ± 0.02 | -0.92** | -0.79** | NA | -0.40** | 15 ± 5 | 0.18 ± 0.07 |
| 5 | -0.52 ± 0.01 | -0.96** | -0.50** | -0.74** | -0.71** | 14 ± 4 | 0.16 ± 0.06 |
| 6 | -0.41 ± 0.02 | -0.75** | -0.60** | 0.20* | 0.40** | 12 ± 3 | 0.13 ± 0.04 |
| 7 | -0.37 ± 0.02 | -0.75** | -0.78** | 0.59** | 0.20* | 13 ± 2 | 0.16 ± 0.03 |
| 8 | -0.21 ± 0.01 | -0.74** | -0.58** | 0.55** | -0.09 | 12 ± 2 | 0.13 ± 0.03 |
| 9 | -0.46 ± 0.03 | -0.76** | -0.63** | 0.04 | -0.46** | 11 ± 4 | 0.13 ± 0.05 |
| 10 | -0.40 ± 0.03 | -0.16 | -0.52** | 0.20 | -0.66** | 10 ± 3 | 0.11 ± 0.03 |
| 11 | -0.28 ± 0.01 | -0.94** | -0.41* | -0.28 | -0.76** | 9 ± 2 | 0.09 ± 0.03 |
| 12 | -0.29 ± 0.02 | -0.41** | -0.61** | 0.17 | 0.18 | 10 ± 2 | 0.11 ± 0.02 |
| 13 | -0.45 ± 0.02 | -0.85** | -0.71** | 0.69** | 0.49* | 10 ± 4 | 0.11 ± 0.05 |
| 14 | -0.18 ± 0.01 | -0.53** | -0.11 | 0.07 | -0.36* | 10 ± 1 | 0.11 ± 0.02 |
| 15 | -0.45 ± 0.02 | -0.57** | -0.63** | 0.87** | 0.45* | 9 ± 2 | 0.10 ± 0.02 |
| 16 | -0.16 ± 0.01 | -0.59** | -0.55** | 0.13 | -0.04 | 9 ± 1 | 0.10 ± 0.02 |
| 17 | -0.46 ± 0.02 | -0.74** | -0.78** | NA | 0.64** | 9 ± 2 | 0.09 ± 0.03 |
| 18 | -0.45 ± 0.01 | -0.82** | -0.81** | 0.65** | 0.76** | 9 ± 2 | 0.09 ± 0.02 |


By contrast, measurements of enhanced nighttime RGM at MBO during summer 2005 showed a slope nearer to unity for RGM versus GEM (-0.89) (Swartzendruber et al., 2006). These events, as well as other instances of enhanced RGM at MBO, were hypothesized to have UT/LS influence (Swartzendruber et al., 2006; Timonen et al., 2013). An aircraft study also reported an $Hg^{II}$ versus $Hg^0$ regression slope for air originating from the upper troposphere that was similarly close to unity (-0.93), and a slope of -0.53 for stratospheric air (Lyman and Jaffe, 2012). The lack of mass closure under stratospheric influence was attributed to the idea that THg decreases toward






the stratosphere while the ratio of $Hg^{II}$:$Hg^0$ simultaneously increases with altitude (Swartzendruber et al., 2006;
Lyman and Jaffe, 2012). While a few slopes of individual events in the present study showed values closer to unity
(e.g. Event 3 = -0.76 ± 0.01), the total slope of all events was much lower than that seen in air masses influenced by
the UT/LS. Alternatively, slopes between -1 and 0 could indicate upwind $Hg^0$ oxidation followed by $Hg^{II}$ loss via
deposition during transport, an explanation which is further supported by the modeled transport behavior during
events at SPL.

      During all high $Hg^{II}$ events at SPL in 2021 and 2022, the air masses generally maintained transport in the
low to mid-free troposphere over the Pacific Ocean before subsiding over the continent (Fig. A2). The back
trajectories from all 18 events spent an average of 12 ± 2% and a maximum of 16% of total transport time in the
PBL. Moreover, HYSPLIT-STILT transport analysis showed that event air masses spent on average just 13 ± 5%
(Fig. 4), and a maximum 24% of transport time above 6 km AGL. The median percent transport time spent in the 3
to 5 km range (chosen here to approximate the low-mid free troposphere) during the events was 43%. The nine air
masses that spent greater than 43% of their transport at these altitudes had significantly (p < 0.001) higher mean $Hg^{II}$
concentrations (142 ± 23 pg m$^{-3}$) and spent significantly less time in the 0 to 2 km range (30 ± 3%) than those that
spent less than 43% of transport in this altitude range (114 ± 11 pg m$^{-3}$; 38 ± 4% of transport in 0 to 2 km altitude
range).  The relatively limited influence from both the UT/LS and PBL on the sampled air masses indicated that
extended periods of elevated $Hg^{II}$ at SPL likely originated in the low to mid-free troposphere (Fig. 4). A comparison
to transport pathways of air masses associated with low $Hg^{II}$ is described in Sect. 3.2.3.

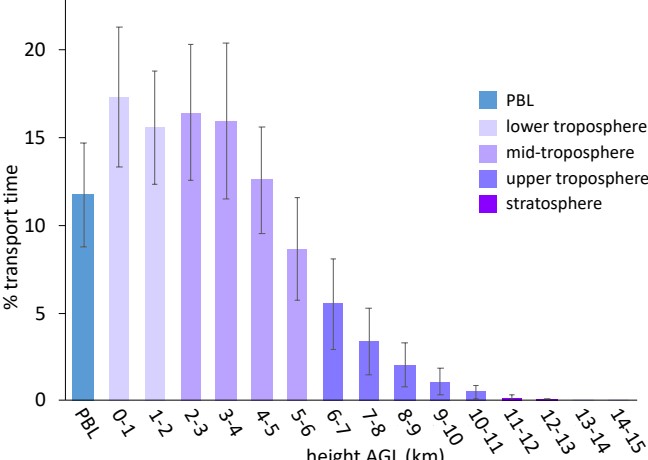

**Figure 4: The percentage of transport time that high $Hg^{II}$ event air masses spent at each altitude above
ground level (AGL) averaged across all 18 events. In general, the majority of air mass transport fell in the low
to mid altitudes, while the air masses spent less of the transport time within the planetary boundary layer**
**(PBL) or in the upper troposphere / lower stratosphere (UT/LS). Bars are color-coded by layers of the
atmosphere. The PBL was explicitly calculated by the HYSPLIT-STILT model, but the other altitude
groupings are general approximations.**



All HYSPLIT-STILT runs demonstrated that SPL experienced prevailing westerlies, as has been demonstrated in earlier work at the site (Faïn et al., 2009; Hallar et al., 2016). However, there was not a consistent

directional origin evident by season (Fig. A2), or in relation to covariance in other air mass tracers (Sect. 3.2.2) during high $Hg^{II}$ events. There was also large variability in whether the air masses originated over the central Pacific, north Pacific, or southwest as well as the horizontal distance covered in 10 days of transport. Although 10 events showed some or all back trajectories approaching SPL from the southwest, there did not appear to be any consistency in air mass speed, altitude, or composition amongst this subgroup. The horizontal transport pathway

alone, therefore, did not appear to directly influence the presence of high concentrations of $Hg^{II}$ at SPL.

There was some variation in the amount of time the simulated air masses spent over the North American continent (50 to 150 hours of the 10-day transport), but differences were not associated with particular air mass compositions. Simulated air mass temperature was most dependent on season, but generally increased over the continent. Additionally, all event back-trajectories showed some upwind precipitation during the 10 days, which

could account for the lack of mass closure between $Hg^0$ and $Hg^{II}$, but there was no precipitation within at least 50 hours of arrival at SPL. Relative humidity remained below 60% during transport, and consistently declined as the air masses approached the site. Therefore, the air masses in which high $Hg^{II}$ was measured at the site were generally dry, and did not experience significant washout close to the measurement site.

In another study at MBO, maximum RGM:GEM ratios ranged from 0.16 to 1.05 during high RGM periods,

with the largest ratios seen in air masses influenced by the marine boundary layer (MBL) that were marked by circulation above the ocean for at least 10 days prior to arrival at MBO as well as decreases in CO, aerosol scattering, and $O_3$ (Timonen et al., 2013). Timonen et al. (2013) defined MBL events as clean air masses that circled above the ocean for at least 10 days prior to arrival at MBO, where enhancements in RGM coincided with decreases in CO, aerosol scattering, and $O_3$, as well as no indication of UT/LS influence. These MBL events were associated

with higher concentrations of RGM and larger RGM/GEM ratios than seen at SPL (Table 3). By comparison, the mean ratio (s.d.) of $Hg^{II}:Hg^0$ during the 18 events at SPL was $0.12 \pm 0.03$, and maximum values ranged from 0.13 to 0.29 (a comparison of these values to non-event times will be discussed in Sect. 3.2.3). Mean ratios during the summer 2021 Events 4–8 ($0.15 \pm 0.02$) were significantly higher than the rest of the events in the study ($0.10 \pm 0.01$) ($p < 0.001$), possibly related to mean $Hg^{II}$ concentrations being significantly higher in summer 2021 than in other

seasons (Sect. 3.1.1). Nevertheless, air masses at SPL did not show behavior or composition comparable to that of MBL-influenced air at MBO. Understandably, by nature of the site locations, the air masses measured at SPL spent much more time over the continent (Fig. A2), potentially allowing for more scavenging of $Hg^{II}$ during transport, and therefore resulting in $Hg^{II}$ vs. $Hg^0$ regression slopes further from unity and a lower ratio of $Hg^{II}:Hg^0$.

In conclusion, based upon the results of the HYSPLIT-STILT and other meteorological analysis, we posit

that the lack of mass closure in the slopes of $Hg^{II}$ versus $Hg^0$ regressions during the 18 events at SPL was likely caused by distant upwind oxidation followed by $Hg^{II}$ loss via deposition during transport. This finding is further supported by the analyses of trace gas measurements and of select non-event periods below.



*3.2.2 Event Air Mass Composition*








Concentrations and relationships between other trace gases measured in this study varied across the events. Of the 16 events with sufficient $O_3$ measurement coverage, seven showed significant positive correlations between $Hg^{II}$ and $O_3$ (Events 3, 6, 7, 8, 13, 15, and 18), and six of these seven events also had significant positive correlations between $Hg^{II}$ and aerosol $PM_1$ scattering (all but Event 8) (Table 3). These events were mostly in summer, but occurred in both study years. Event 3 is shown in the Appendix as an example event where $Hg^{II}$ was positively correlated with $O_3$ and $PM_1$ $\sigma_{sp}$ (Fig. A3). Ozone concentrations increased by approximately 10 to 20 ppbv during these events, concurrent with fluctuations in $Hg^{II}$ concentrations. Meanwhile, the magnitudes of aerosol $PM_1$ scattering enhancements were more varied, ranging from approximately 30 to 120 $Mm^{-1}$; in part because scattering was generally lower during spring than summer, and some of the summer events also showed evidence of smoke (Sect. 2.3.1).

While Faïn et al. (2009) reported similar transport altitudes for air masses associated with enhanced $Hg^{II}$ at SPL, their study found no relationship between $Hg^{II}$ and $O_3$. Previous studies at the MBO and PDM mountaintop sites, however, have also shown co-enhancements of $Hg^{II}$ and $O_3$. Often, these enhancements occurred with simultaneous decreases in $Hg^0$, CO, and aerosol scattering, which Swartzendruber et al. (2006), Timonen et al. (2013), and Fu et al. (2016) attributed to $Hg^{II}$ transport from the UT/LS. As previously discussed, however, HYSPLIT-STILT transport analysis during events at SPL did not show air masses at high enough altitudes to indicate UT/LS influence in the 10 days of simulated transport (Fig. A2).

High $O_3$ concentrations could also be the result of multiple forms of combustion processes, both natural and anthropogenic. Timonen et al. (2013) also reported events of enhanced $Hg^{II}$, $O_3$, aerosol scattering, and CO at MBO during springtime. Seasonal spring meteorological conditions favor long-range trans-Pacific air mass transport and can deliver anthropogenic pollution from the Asian continent (Timonen et al., 2013). Maximum $O_3$ concentrations during the seven events at SPL ranged from 58 to 74 ppbv, values which were comparable to the maximum $O_3$ values reported at MBO during events of high $Hg^{II}$ associated with influence from Asian long-range transport, which ranged from 69 to 77 ppbv (Timonen et al., 2013). However, all of the events in this study with positive correlations between $Hg^{II}$ and $O_3$ occurred during summertime, with the exception of Event 3 during spring 2021. In all cases except for Event 8, back trajectories did not show transport from the Asian continent in the 10-day histories (Fig. A2). Alternatively, $O_3$ could have been picked up from the North American PBL as the air masses traveled over the continent before arriving at SPL.

The enhancement of $O_3$ during some summer events could also be a result of biomass burning. Eight of the 18 events occurred when SPL was in smoke from local or regional wildfires and experiencing elevated concentrations of combustion tracers, particularly CO and aerosol scattering (Sect. 2.3.1, Table 2). Both $O_3$ and CO were significantly positively correlated with $Hg^{II}$ during three events (Events 13, 15, 18), and CO was above 150 ppbv during five events when $Hg^{II}$ was significantly correlated with $O_3$ (Events 6, 7, 8, 13, 18). Ozone can be a secondary product in smoke plumes of wildfires (Briggs et al., 2016), and four events where $Hg^{II}$ was correlated with $O_3$ also showed positive correlations between $O_3$ and CO (Events 6–8, 18). Biomass burning has also been shown to volatilize stored Hg as $Hg^0$ and release it to the atmosphere. High levels of $Hg^{II}$ have not typically been



reported in smoke plumes (McLagan et al., 2021), though all available measurements were collected with inaccurate methods. Elevated $Hg^{II}$ concentrations during the events under consideration here, which also had concentrations of $Hg^0$ comparable to the seasonal means, may be coincidental to the presence of smoke in the air masses; i.e., it is possible that the smoky air was mixed with cleaner free tropospheric air containing higher concentrations of $Hg^{II}$.

Seven events showed significant anticorrelations between $Hg^{II}$ and $PM_1$ scattering (Events 1, 4, 5, 9, 10, 11,
14; Table 3). Oxidized Hg in these events also tended to be significantly anticorrelated with $NO_x$, but not always with other pollution tracers. Three of these events occurred when SPL was in smoke, and therefore $PM_1$ scattering was elevated, but showed measurable decreases when $Hg^{II}$ increased, indicating that SPL may have experienced some influence from cleaner, free tropospheric air during these smoky periods. Event 5 (June 13 23:00 – June 15, 2021 21:00 MST) (Fig. A4) was the only one of these cases where $Hg^{II}$ was significantly anticorrelated with $O_3$
(Pearson's R = -0.74, p < 0.001), and was also anticorrelated with $NO_x$ (Pearson's R = -0.43, p < 0.001) and $SO_2$ (Pearson's R = -0.34, p < 0.05). This event also had a particularly strong anticorrelation between $Hg^{II}$ and $Hg^0$ (Pearson's R = -0.96, p < 0.001) and RH (Pearson's R = -0.50, p < 0.001), and notably higher-altitude transport than other events (Fig. A4); the air mass spent 23% of transport time higher than 6 km AGL, compared to an average of $13 \pm 5\%$ for all events. Considering these factors, Event 5 appeared to have particularly clean air conditions,
originating from higher in the free troposphere. However, the altitudes associated with Event 5 air mass transport were still not high enough to be considered UT/LS influenced, as the air mass trajectories spent 93% of transport time below 8 km AGL.

The occurrence of events with either positive or negative relationships with aerosol scattering could be related to the relative abundance of gaseous versus particulate $Hg^{II}$ (e.g. RGM/GOM vs. PBM). For example, an
enhancement in aerosol scattering could indicate a higher proportion of PBM than during other events of elevated $Hg^{II}$, whereas events with lower aerosol scattering could indicate more gaseous $Hg^{II}$. However, a study at PDM showed that during events of elevated PBM, measured with the Tekran speciation system, aerosol number concentration was significantly anticorrelated with PBM (Fu et al., 2016). The authors posited that this relationship could indicate that atmospheric aerosol concentration may not play a significant role in PBM formation in the
middle and upper free troposphere, or rather that aerosol number concentration at PDM is driven by anthropogenic influence from the PBL, and is therefore not representative of the composition of the middle to upper free troposphere at that site (Fu et al., 2016). Because the dual-channel Hg measurement system does not differentiate between gaseous and particle phases of $Hg^{II}$, any relationship between these variables is speculative at this time.

*3.2.3 June 2022 case study*

June 2022 contained five distinct events of elevated $Hg^{II}$ (Events 11–15), with periods of low $Hg^{II}$ in between events (referred to here as "non-events" and numbered 1 through 5) (Fig. 5). The five events during this period had characteristics similar to the other 13 events from this study, such as depleted $Hg^0$ and low RH, but displayed variation in transport pathways, meteorology, and trace gas concentrations. More specifically, all the June 2022 events had strong and significant anticorrelations between $Hg^{II}$ and $Hg^0$ (Pearson's R = -0.94 to -0.41, p <



0.001; Table 3), whereas the non-event Hg$^{II}$ versus Hg$^0$ anticorrelations were either weaker (Non-Events 1, 3, and 5)
       or not significant at the $p < 0.05$ level (Non-Events 2 and 4) (Table A2).

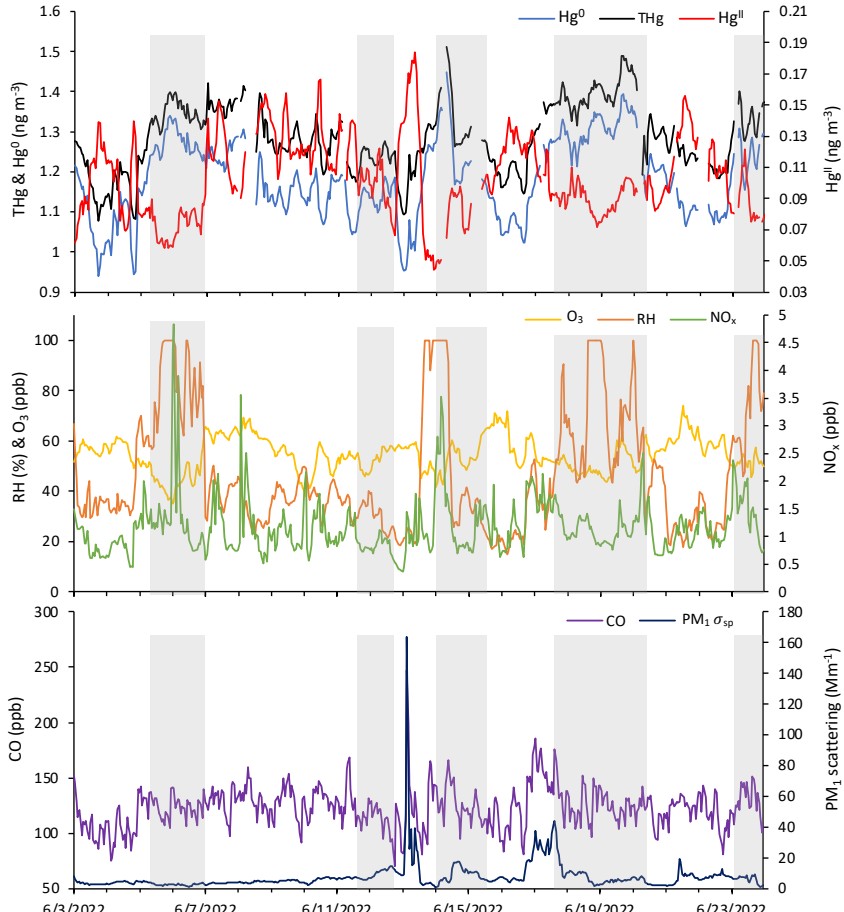

**Figure 5: Time series of (a) Hg$^{II}$, Hg$^0$, THg, (b) RH, O$_3$, NO$_x$, and (c) CO, and aerosol PM$_1$ scattering for June**
**3 04:00–June 24, 2022 03:00 MST, showing the five events of enhanced Hg$^{II}$ that occurred in June 2022, and**
**the five corresponding non-events (shaded in gray).**

       The non-event air masses spent a significantly smaller percentage of transport time in the 3 to 5 km altitude
       range ($26 \pm 4\%$) than the event air masses ($40 \pm 7\%$) ($p < 0.05$) (Fig. A5). Generally, event air masses tended to
       spend more time at altitudes associated with the low to mid-free troposphere, whereas non-event air masses spent
more time at lower altitudes and in the PBL (Fig. A5 and A6). June event air masses spent 52% of transport time
       above 3 km, whereas non-events spent 44% of transport time above 3 km. Additionally, the high Hg$^{II}$ events showed
       significantly higher mean Hg$^{II}$ and O$_3$ concentrations, and significantly lower mean Hg$^0$, RH, and NO$_x$
       concentrations than the non-events (Table 4). The ratio of Hg$^{II}$:Hg$^0$ was also significantly higher ($p < 0.05$) during



events (10 ± 1%) than non-events (6 ± 1%), and the amount of THg measured as Hg$^{II}$ was also significantly (p < 0.001) higher during events (10 ± 1%) than non-events (7 ± 1%). Simulated precipitation for the air masses prior to arrival at SPL showed precipitation up to the simulation end time in Non-Events 3, 4, and 5. The higher RH during non-event times concurrent with lower Hg$^{II}$ concentrations could indicate wet deposition of Hg$^{II}$. Events 13 and 14 occurred when SPL was influenced by smoke from regional wildfires in Arizona and New Mexico, so both CO and aerosol scattering were elevated (Table 2). Figure 6 shows example transport models for Event 11 (June 3 04:00 – June 5, 2022 04:00 MST) and Non-Event 1 (June 5 05:00 – June 6, 2022 23:00 MST). Event 11 showed higher transport altitudes, lower RH, and no precipitation directly prior to arrival at SPL, whereas Non-Event 1 had lower atmospheric transport, higher RH, and more recent precipitation. The differences between events and non-event periods demonstrated that the commonalities in event air mass composition and transport could be attributed to the specific conditions under which Hg oxidation occurred in the upwind atmosphere, as opposed to ambient atmospheric conditions seen locally at SPL.

**Table 4: June 2022 event vs non-event mean ± s.d. for Hg species, THg, RH, PM$_1$ σ$_{sp}$, and trace gases. Bolded values are significantly different at the p < 0.05 level.**

|  | Hg$^{II}$ (pg m$^{-3}$) | Hg$^{0}$ (ng m$^{3}$) | THg (ng m$^{3}$) | RH (%) | NO$_x$ (ppb) | O$_3$ (ppb) | CO (ppb) | PM$_1$ σ$_{sp}$ (Mm$^{-1}$) |
|---|---|---|---|---|---|---|---|---|
| Events | **112 ± 23** | **1.1 ± 0.1** | 1.3 ± 0.1 | **37 ± 15** | **1.1 ± 0.4** | **57 ± 6** | 120 ± 20 | 6.7 ± 5.4 |
| Non-Events | **85 ± 16** | **1.3 ± 0.1** | 1.4 ± 0.1 | **65 ± 26** | **1.3 ± 0.6** | **49 ± 6** | 130 ± 20 | 7.8 ± 7.1 |



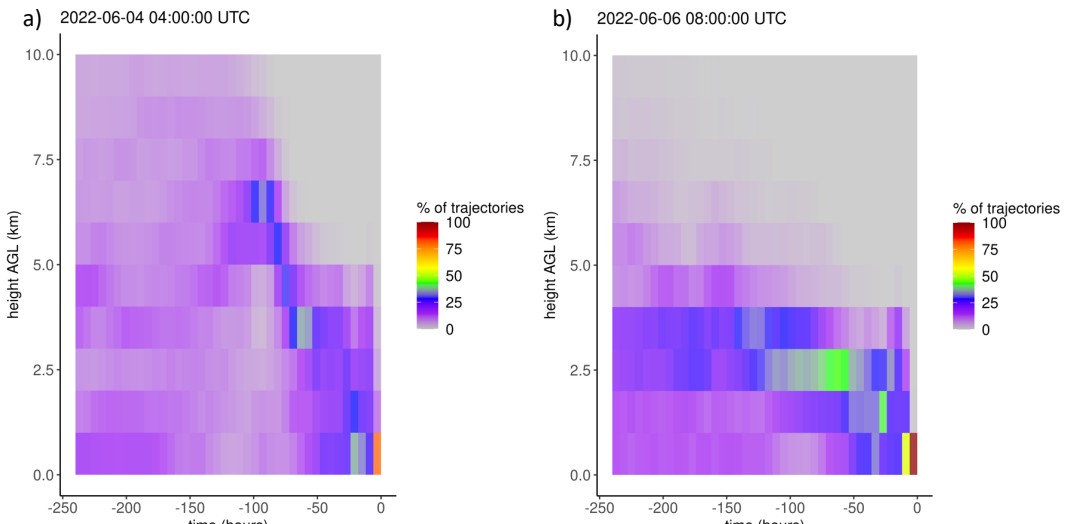

**Figure 6:** Example vertical distributions of air mass trajectories for (a) Event 11 and (b) Non-Event 1. Shading
reflects the percentage of backward trajectories within a given altitude bin when aggregated to 6-hour intervals.

*3.3 Principal Components Factor Analysis (PCA)*

Broadening these event-based analyses to the rest of the hourly measurement data, three factors explaining
60–70% of the total variance were generated in the PCA data reduction technique applied to each of the sampled
seasons in 2021 and 2022 datasets (Table B1 and B2), and for the combined 2022 spring–summer period (Table 5).
Although differences appeared in the magnitude and sign of variable loadings for each period, there were some
notable features that all simulations had in common, as well as some consistent seasonal patterns. For example, all
solutions contained two common factors: one representing clean background air with evidence of Hg oxidation
(little to no loading of most combustion tracers, and inverse loadings of $Hg^0$ and $Hg^{II}$), and another representing
anthropogenic and/or biogenic combustion (with some combination of CO, $NO_x$, aerosol scattering, $O_3$ and $Hg^0$
loading with the same sign). In summer 2021 this factor explained the largest percent variance (32%) and was likely
dominated by the aforementioned biomass burning signature, whereas in summer 2022 it explained just 23% of the
variance and likely represented other local regional combustion sources (Tables B1-B2). The third factor varied in
its makeup, but in at least the 2022 applications there were shared loadings of pressure, water vapor mixing ratio and
aerosol scattering, and an inverse loading of CO. Another consistent feature was that $Hg^{II}$ almost exclusively loaded
inversely with $Hg^0$, and on any factor where the two variables did display the same sign, one or both of their
loadings were very small (< 0.3). Oxidized Hg usually, but not always, inversely related to water vapor mixing ratio,
and in some solutions was positively associated with $O_3$ and aerosol scattering but did not load strongly with factors
representing combustion sources. These broader features are complimentary with the relationships observed during
events of high $Hg^{II}$ (Sect. 3.2).









**Table 5: Factor loadings of each variable and the percentage of total variance explained by each factor, as obtained from Principal Components Analysis for the spring-summer 2022 period.**

| | 2022 - All | | |
|---|---|---|---|
| | Factor 1 | Factor 2 | Factor 3 |
| pressure | 0.89 | 0.17 | -0.08 |
| wvmr | 0.86 | -0.22 | 0.04 |
| $Hg^0$ | 0.12 | -0.66 | 0.51 |
| $Hg^{II}$ | 0.11 | 0.78 | -0.20 |
| CO | -0.42 | -0.04 | 0.76 |
| $O_3$ | -0.02 | 0.72 | 0.17 |
| NOx | 0.25 | 0.00 | 0.77 |
| $PM_1\ \sigma_{sp}$ | 0.59 | 0.53 | 0.26 |
| Variance Explained | 30% | 24% | 17% |

Here we describe the results of PCA application to the combined period of 1 March – 15 September, 2022 (Table 5); summaries of the individual spring and summer seasons are provided in the Appendix. Factor 1 (30%)

displays strong positive loadings of pressure, water vapor, and aerosol scattering with a moderate negative loading of CO. This factor may represent particle climatology given SPL's history of spending a significant fraction of time in-cloud and evidence for new particle formation during particular seasons and times of day (Hallar et al., 2016). Factor 2 (24% of variance) reflects observations of Hg oxidation within the clean, dry, remote free troposphere. Interestingly, both ozone and aerosol scattering loaded strongly with the same sign as $Hg^{II}$ on this factor, a feature

that also appeared to some degree in spring 2021 and in the separate analyses of spring 2022 and summer 2022. As shown in Sect. 3.2.2, about one third of the 18 high $Hg^{II}$ events also had positive correlations for $Hg^{II}$ versus $O_3$ and $Hg^{II}$ versus aerosol scattering. These relationships may point to the potential for Hg oxidation in the presence of ozone, but not necessarily in air originating from the UT/LS. This may also suggest the propensity for $Hg^{II}$ to be found in the particulate form at SPL in certain instances, but we cannot confirm this due to the current inability of

the dual-channel system to distinguish between phases of $Hg^{II}$ (e.g. GOM or PBM). Lastly, Factor 3 in 2022 (17% of variance) represents combustion sources with positive loadings of CO, $NO_x$, and $Hg^0$ with more moderate loadings of $O_3$ and aerosol scattering. It is notable that in the full 2022 analysis, aerosol scattering distributed almost uniformly across all three factors, suggesting multiple drivers for the presence of aerosols at the site that likely vary by season, as evident in the separate spring and summer analyses.

**4. Conclusions**

In this study, we examined air mass composition and transport of events of elevated $Hg^{II}$ at SPL, a high elevation mountaintop site, over two six-month periods in spring and summer 2021 and 2022. Unlike previous





studies at mountaintop sites, we employed a dual-channel Hg measurement system, which was calibrated with an International System of Units (SI)-traceable calibration system and shown to produce unbiased measurements of

$Hg^{II}$. Elemental Hg concentrations and trends at SPL were similar to those reported in previous work, but mean and maximum $Hg^{II}$ concentrations in this study were approximately three times higher than earlier measurements at this site, and $Hg^{II}$ comprised on average more than 10% of total atmospheric Hg during high $Hg^{II}$ events. We also demonstrated that Hg concentrations at SPL were not affected by emissions from the three upwind coal-fired power plants, which can likely be attributed to power plant emission controls and lower Hg content in western coal. Events

of elevated $Hg^{II}$ showed evidence of upwind $Hg^{0}$ oxidation, followed by $Hg^{II}$ loss during transport in the low to mid-free troposphere and no evidence of UT/LS influence. PCA confirmed that $Hg^{II}$ measured at SPL was a result of Hg oxidation in the background atmosphere.

Results from this study contribute to the current understanding of Hg oxidation in a remote continental atmosphere. Additionally, the implementation of the dual-channel system provided Hg measurements that were

larger in magnitude and more accurate than commercially available instrumentation. Concurrent work related to this project at SPL further elaborates on the methodological improvements for measuring ambient $Hg^{0}$ and $Hg^{II}$ (Elgiar et al., in review), the potential contribution of iodine as an emerging Hg oxidant (Lee et al., in review), and the effect of washout on ambient $Hg^{II}$ concentrations (Weiss-Penzias et al., in review). Collectively, the results of this campaign will importantly advance the current understanding of ambient Hg origins, cycling, bioavailability, and

ultimately ecosystem fate.




**Appendices**

*Appendix A. Events of High Oxidized Mercury*

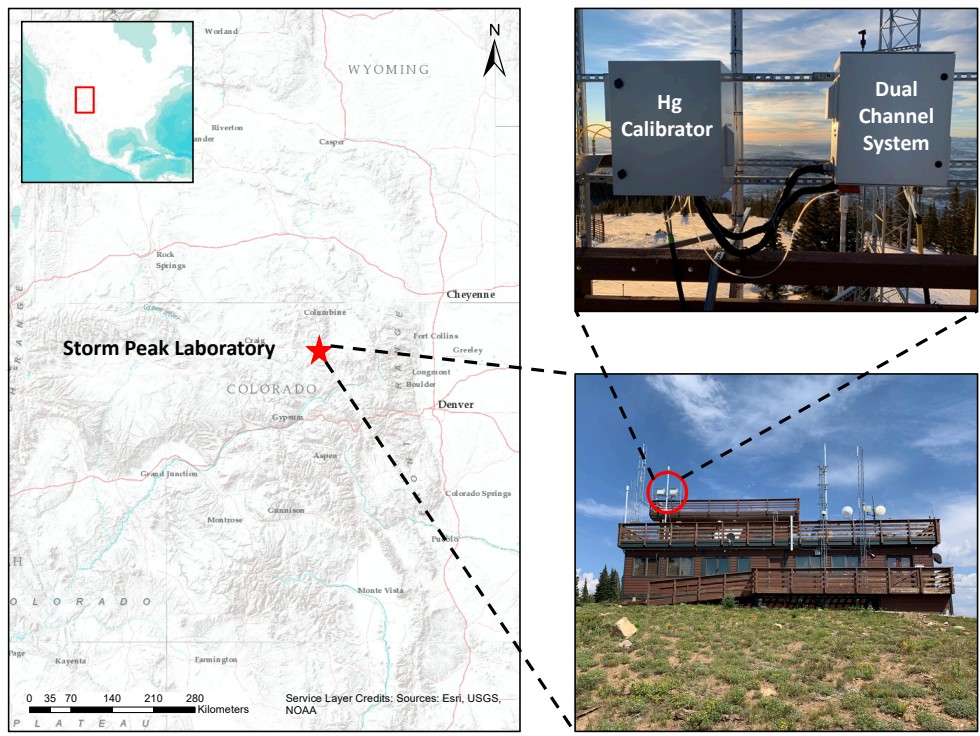

**Figure A1: Site map of Storm Peak Laboratory and the location of the dual-channel system on the roof of the laboratory (Elgiar et al., in review).**




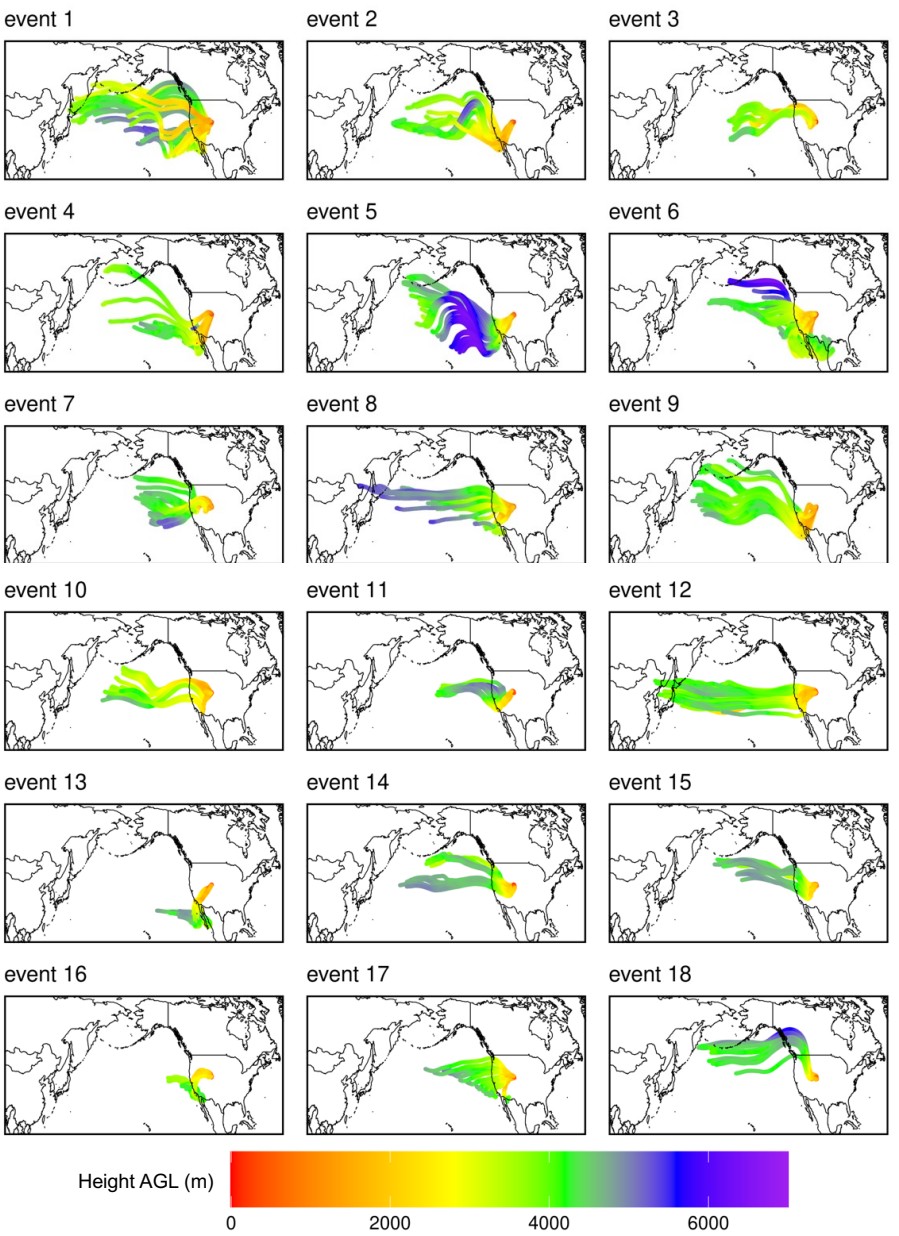

**Figure A2: Composite maps of averaged HYSPLIT-STILT 10-day back trajectories for all 18 events of**
**elevated Hg$^{II}$, color-coded by altitude AGL. Depicted trajectories represent the simulation average transport**
**pathway of 1,000 backward trajectories, with simulations initialized at 3-hour intervals throughout each**
**event.**





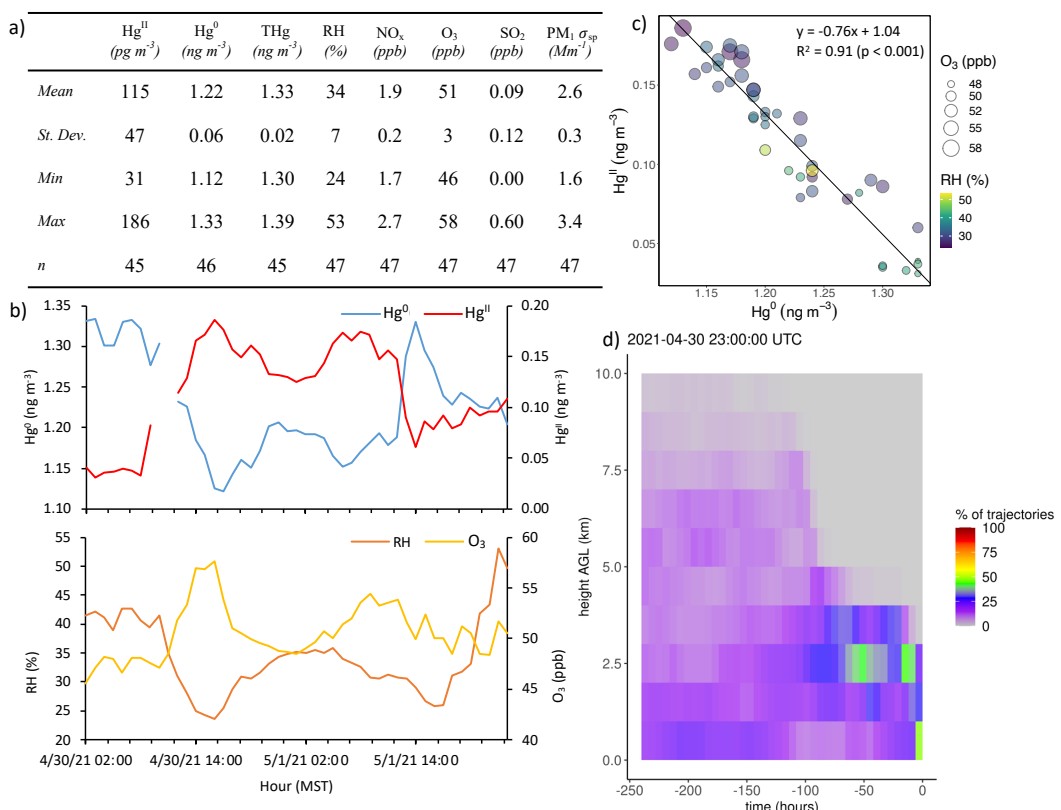

Figure A3: Event 3 (a) descriptive statistics, (b) time series of Hg$^0$, Hg$^{II}$, RH, and O$_3$, (c) scatterplot of Hg$^{II}$ vs Hg$^0$ with RH and O$_3$, and (d) an example vertical profile of HYSPLIT-STILT backward trajectories, with the average mixed layer height represented by a black line.



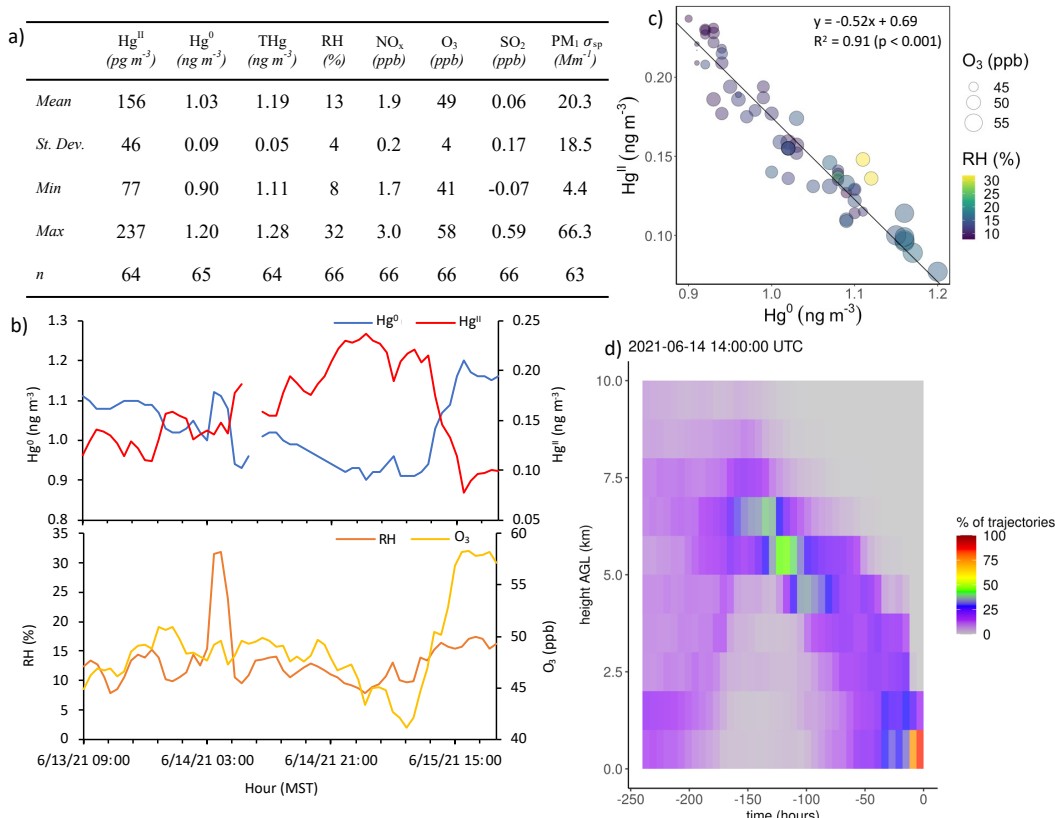

**Figure A4: Event 5 (a) descriptive statistics, (b) time series of Hg[0], Hg[II], RH, and O₃, (c) scatterplot of Hg[II] vs Hg[0] with RH and O₃, and (d) an example vertical profile of HYSPLIT-STILT backward trajectories, with the average mixed layer height represented by a black line.**



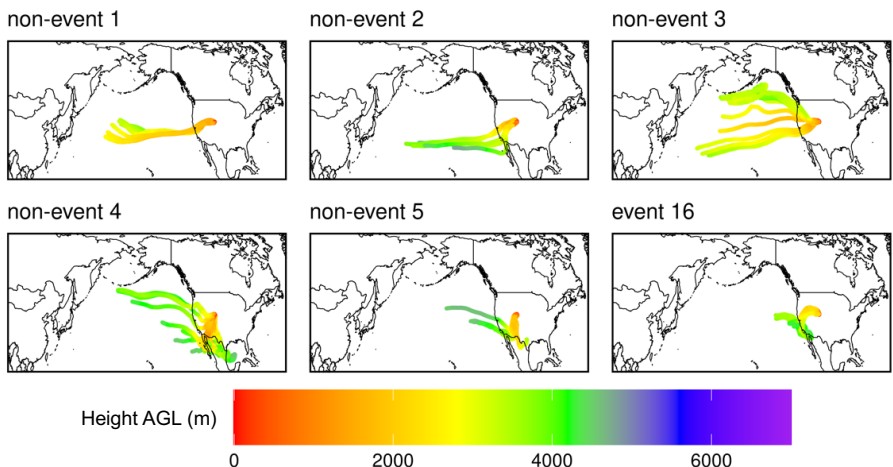

**Figure A5: Composite maps of averaged HYSPLIT-STILT 10-day back trajectories for the five Non-Events in the June 2022 case study.**

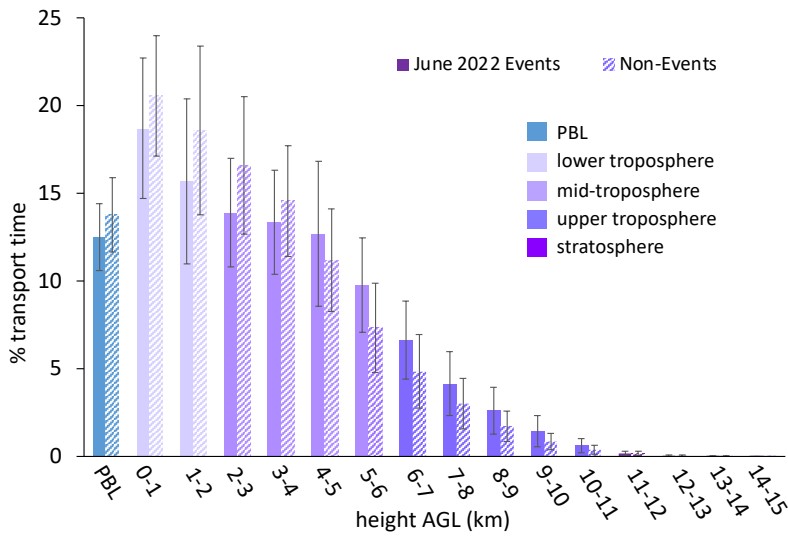

**Figure A6: The percentage of transport time that high $Hg^{II}$ event air masses spent at each altitude above ground level (AGL) averaged for the five June 2022 events and the corresponding June 2022 non-events. Event air masses tended to spend more time at altitudes associated with the low to mid-free troposphere, whereas non-event air masses spent more time at lower altitudes and in the PBL. Bars are color-coded by**



layers of the atmosphere. The PBL was explicitly calculated by the HYSPLIT-STILT model, but the other altitude groupings are general approximations.


Table A1: Pearson's correlation coefficients and p-values for $Hg^{II}$ versus water vapor mixing ratio (wvmr) for all data in the study period, and for all times when relative humidity was below 85%, to limit the analysis to outside of periods when SPL was in cloud. While $Hg^{II}$ and wvmr were generally significantly anticorrelated, making it a strong variable for PCA, but RH had the most robust relationship with $Hg^{II}$, similar to previous

studies at SPL (Faïn et al., 2009). Significance is defined as *p < 0.05, **p < 0.001.

| $Hg^{II}$ v.s. wvmr | All data | RH < 85% |
|---|---|---|
| Spring 2021 | -0.12** | -0.04 |
| Summer 2021 | -0.54** | -0.54** |
| Spring 2022 | 0.05 | 0.17** |
| Summer 2022 | -0.53** | -0.50** |
| All data | -0.12** | -0.24** |

Table A2: Pearson correlation coefficients for the five Non-Events from the June 2022 case study. *p < 0.05,

**p < 0.001.

| $Hg^{II}$ v.s. | $Hg^0$ | RH | $O_3$ | CO | $PM_1\ \sigma_{sp}$ |
|---|---|---|---|---|---|
| Non-Event 1 | -0.45* | -0.57** | 0.51* | 0.13 | -0.07 |
| Non-Event 2 | -0.13 | -0.32 | 0.13 | -0.14 | 0.49* |
| Non-Event 3 | -0.50* | -0.83** | 0.87** | -0.34 | 0.56* |
| Non-Event 4 | -0.13 | -0.42** | 0.46** | -0.19 | 0.24 |
| Non-Event 5 | -0.59* | -0.63* | 0.25 | 0.46* | 0.42 |



*Appendix B. Principal Components Analysis*

In both spring periods, Factor 1 (27–28% of total variance) was most notably marked by large inverse
loadings of $Hg^0$ and $Hg^{II}$. In spring 2021, Factor 1 also had strong loadings of pressure and aerosol scattering of the
same sign with $Hg^{II}$, whereas in spring 2022 there were strong loadings of CO and $NO_x$ with the opposite sign to
$Hg^{II}$. Notably, both $Hg^0$ and $Hg^{II}$ distributed strongly on Factors 2 and/or 3 during the spring seasons, but always
inversely with one another. For $Hg^{II}$, this may be related to the two different temporal patterns driving $Hg^{II}$
concentrations in the observations, with a strong diel cycle of high daytime $Hg^{II}$ in the early spring (Sect. 3.1) along
with the multi-day episodes of high $Hg^{II}$ seen throughout the study period (Sect. 3.2). Meanwhile, $Hg^0$ also appeared
to associate with the same sign as combustion tracers such as CO, $NO_x$, and/or $O_3$ as well as with water vapor
mixing ratio (Table B1).

The two summer periods showed more consistency between one another in terms of the distributions of
variables across different factors. In summer 2021, Factor 1 (32% of total variance) appeared to represent
combustion with strong loadings of the same sign for CO, $O_3$, $NO_x$, and aerosol scattering and a weaker loading of
$Hg^0$. A similar factor appeared in summer 2022, but as Factor 2 (23% of total variance) and with a weaker loading of
ozone compared to 2021. The previously mentioned strong influence of local or regional wildfire smoke during at
least one third of the summer 2021 period is likely driving the composition of Factor 1 in summer 2021, while the
reduced presence of underlying smoke in summer 2022 but still the existence of other regional combustion sources
likely explains the presence of the combustion fingerprint as Factor 2 in summer 2022. Meanwhile, Factor 2 in 2021
(27% of variance) and Factor 1 in 2022 (33% of variance) are consistent with the proposed Hg oxidation in the
background dry free troposphere, as indicated by a strong inverse loading of $Hg^{II}$ with $Hg^0$ and also a strong inverse
loading of $Hg^{II}$ with water vapor mixing ratio, as well as very weak or inverse loadings of combustion tracers such
as CO or $NO_x$. Interestingly, in summer 2022 this factor also had a strong loading of ozone and a weak loading of
aerosol scattering both with the same sign as $Hg^{II}$, but this was much less evident in summer 2021 (Table B2).

**Table B1: Factor loadings of each variable and the percentage of total variance explained by each factor, as
obtained from Principal Components Analysis for the spring seasons of 2021 and 2022.**

| | 2021 | | | 2022 | | |
|---|---|---|---|---|---|---|
| | Factor 1 | Factor 2 | Factor 3 | Factor 1 | Factor 2 | Factor 3 |
| pressure | 0.43 | -0.22 | -0.12 | -0.09 | 0.30 | 0.66 |
| wvmr | -0.18 | -0.11 | 0.81 | 0.09 | -0.22 | 0.76 |
| $Hg^0$ | -0.39 | 0.27 | 0.70 | 0.75 | -0.35 | 0.15 |
| $Hg^{II}$ | 0.73 | -0.11 | -0.14 | -0.45 | 0.52 | 0.37 |
| CO | -- | -- | -- | 0.72 | 0.08 | -0.36 |
| $O_3$ | 0.22 | -0.76 | 0.27 | 0.04 | 0.69 | -0.19 |
| NOx | 0.09 | 0.84 | 0.27 | 0.72 | 0.16 | 0.11 |
| $PM_1\ \sigma_{sp}$ | 0.80 | 0.17 | 0.02 | 0.02 | 0.80 | 0.17 |
| Variance Explained | 28% | 18% | 15% | 27% | 18% | 16% |



**Table B2: Factor loadings of each variable and the percentage of total variance explained by each factor, as obtained from Principal Components Analysis for the summer seasons of 2021 and 2022.**

|  | 2021 | | | 2022 | | |
|---|---|---|---|---|---|---|
|  | Factor 1 | Factor 2 | Factor 3 | Factor 1 | Factor 2 | Factor 3 |
| pressure | -0.09 | 0.06 | 0.92 | 0.06 | 0.05 | 0.94 |
| wvmr | -0.21 | 0.85 | 0.01 | 0.82 | 0.24 | 0.20 |
| $Hg^0$ | 0.28 | 0.84 | 0.06 | 0.78 | 0.31 | 0.03 |
| $Hg^{II}$ | 0.11 | -0.81 | 0.09 | -0.83 | 0.06 | 0.01 |
| CO | 0.89 | -0.15 | -0.19 | 0.11 | 0.79 | -0.39 |
| $O_3$ | 0.51 | -0.16 | 0.62 | -0.67 | 0.23 | 0.18 |
| NOx | 0.67 | 0.29 | 0.11 | 0.19 | 0.79 | 0.14 |
| $PM_1$ $\sigma_{sp}$ | 0.87 | -0.15 | 0.17 | -0.25 | 0.66 | 0.36 |
| Variance Explained | 32% | 27% | 16% | 33% | 23% | 15% |

**Data availability**

 Ambient air data collected during this project is publicly available (Gratz et al., 2024).

**Author contribution**

LG, SL, AGH, and RV planned the campaign; TE, SL, LG, AGH, and NSH collected the measurements; SL and TE developed and improved the dual-channel Hg measurement system; ED, LG, TE, and NWH analyzed the data; JCL developed and TYW ran the HYSPLIT-STILT model; ED and LG wrote the manuscript; SL, AGH, RV, TYW, CFL, PWP, NWH, JCL, TE, and NSH reviewed and edited the manuscript.

**Competing Interests**

The authors declare that they have no conflict of interest.

**Acknowledgements**

Funding for this work was provided by the National Science Foundation (NSF) Awards 1951513, 1951514, 1951515, 1951632. The authors thank Dr. Ian McCubbin, Dan Gilchrist, and Dr. Maria Garcia for assisting with
instrument maintenance and data acquisition, Dr. Betsy Andrews of NOAA ESRL/GML for providing aerosol data, and Megan Ostlie for working up the trace gas data. Colorado College undergraduate students Brandon Chan and Zoe Zwecker contributed to the preliminary data analyses related to the results presented in this manuscript.



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
