# Peer review of "Elevated oxidized mercury in the free troposphere: Analytical advances and application at a remote continental mountaintop site"

_EGUsphere, 2024_

## Author Comment (AC1)

**Reviewer 1:**

General Comments:

This study investigates the levels of oxidized mercury at the high-elevation measurement station Storm Peak Laboratory. The use of a dual-channel system reduces the systematic biases of previous oxidized mercury measurements, and the length of the dataset (2 six-month periods) allows for analysis of drivers of oxidized mercury dynamics. The study is written in a clear fashion and the conclusions are well supported by the statistical analysis. I appreciate that the authors used multiple approaches to analyze their data, including identification and analysis of anomalous events, back-trajectory analysis, and Principle Component Analysis. Not all aspects of the oxidized mercury observations can be fully explained, but this is reasonable given that very few (if any) such datasets exist and this can act as an impetus for further observational and modelling work. I believe that this study is ready for publication and provide only minor comments and questions.

Specific Comments:

-In the schematic, shouldn't fires and anthropogenic sources be emitting Hg(II) as well? I understand that the authors conclude that this does not affect the measurements at SPL, but perhaps should be included for completeness of the cycle.

We thank the reviewer for this suggestion. Although the intent of the schematic is not to show the entirety of the Hg biogeochemical cycle, but rather the specific processes discussed in this manuscript, we changed the diagram to simply "Hg" for both the power plant and biomass burning plumes. There is no current evidence of $Hg^{II}$ being emitted directly from biomass burning plumes (McLagan et al., 2021; Obrist et al., 2007; Friedli et al., 2003); however, it is an ongoing area of research, but not the main focus of this manuscript. Thus, we chose to generalize the emissions from the boundary layer to just "Hg" in order to acknowledge that anthropogenic pollution sources produce both $Hg^0$ and $Hg^{II}$, but we cannot fully explain those processes at this time, nor do our measurements capture the relative emissions of or cycling between these species.

-L67 - couldn't the oxidation mechanisms by Br and OH also occur in the boundary layer, not only the free troposphere? It might be that we have more evidence from the free troposphere, where measurements are not impacted by local emissions, but oxidation could occur throughout the troposphere (and stratosphere - Saiz-Lopez et al., doi:10.1029/2022GL097953, 2022)

We thank the reviewer for bringing up this point. We have edited this section to speak more broadly to Hg oxidation throughout the atmosphere "Previous studies have indicated multiple possible major oxidants of Hg in the atmosphere. While Hg oxidation can occur in the stratosphere, driven by a photosensitized oxidation mechanism (Saiz-Lopez et al., 2022), as well as in both the marine and continental boundary layers (Lyman et al., 2020a), recent studies have suggested that Hg oxidation

occurs primarily in the free troposphere, and the leading oxidants are halogens such as atomic bromine (Br) and the hydroxyl radical (OH) (Dibble et al., 2020)," (L81).

-L371 - I know this is pervasive in mercury literature, but reporting a trend in % per year implies an exponential fit to me. However, all of the cited literature studies calculate linear regression trends. I think it would therefore be preferable to describe these results in ng m-3 yr-1 or % changes over the number of years studied. Could the authors find a way to report their trend results and compare with existing literature without continuing the use of these misleading units?

We have included the results in ng $m^{-3}$ $y^{-1}$ as follows: "Considering this drop of ~0.3 ng $m^{-3}$ over 14 years, it can be estimated that the more recent measurements were lower by 0.021 ng $m^{-3}$ $y^{-1}$ (~1.4% per year) values that are within the range of reported downward trends in northern hemisphere background concentrations (Slemr et al., 2011; Weigelt et al., 2015; Lyman et al., 2020a; Sonke et al., 2023)." (L422). We also added additional values for the studies cited in this section to report values in ng $m^{-3}$ $y^{-1}$ when possible.

-L420 - Is it possible that local photochemistry and deposition dynamics could cause the diurnal pattern of HgII?

The reviewer's point is possible, however given the data available in this study, we are unable to conclusively determine the cause of the springtime diel $Hg^{II}$ pattern. We did add further discussion of our analysis to this section to address possible causes of the diel cycle, including discussion of the temporal behavior in several other potential boundary layer tracers (L475).

-L488 - A useful reference here may be Fu et al., ES&T, 2021, https://doi.org/10.1021/acs.est.1c02568, which also found a similar slope as the current study (-0.44), and a similar origin in the mid-troposphere (5 km)

We thank the reviewer for bringing this study to our attention. The comparison is useful; however, we also noticed that the Fu et al. (2021) study calculated a regression of all the data pre-, post-, and during the free tropospheric intrusion, and corrected their data to account for longitudinal bias. In our model runs for events of elevated $Hg^{II}$ and non-event periods, we did not see as much longitudinal variability as was seen at PDM, thus we did not apply this correction factor. Nevertheless, the characteristics of the free tropospheric intrusion observed in one event by Fu et al. (2021) are similar to those of many of the events of elevated $Hg^{II}$ in this study. We incorporated it into our discussion as follows: "Meanwhile, Fu et al. (2021) found a slope of -0.44 ± 0.10 for $Hg^{II}$ versus $Hg^{0}$ during one event of free tropospheric air mass intrusion at PDM, similar to the mean slope we report here. Unlike the majority of events we identified at SPL, they reported a lack of common air mass origins across their eight-day event and after correcting for this based on reported latitudinal GEM differences, they obtained a corrected slope of -0.88 implying that most of the $Hg^{II}$ was retained within the sampled air masses. Given the relative consistencies in air mass origins (Figure A3) during our events, the majority of which were 5 days or fewer (Table 2), we did not attempt such a correction." (L561).

-L561 - Correct this, ozone is not directly formed by combustion but rather through chemistry of ozone precursors (not all of which are produced by combustion)

We clarified this point to say that "High $O_3$ concentrations could also be generated as a secondary product of natural or anthropogenic combustion," (L657).

-L582 - It's also possible that co-emitted halogens or aerosols in smoke plumes lead to faster oxidation or scavenging of Hg(0) than clean air, leading to elevated Hg(II) and particulate Hg.

We thank the reviewer for this point, and we have included it as follows: "However, it is also possible that co-emitted halogens or aerosols in smoke plumes led to faster oxidation or scavenging of $Hg^0$ than in clean air, leading to elevated $Hg^{II}$ and PBM," (L682).

-L602 - Would aerosol number concentration be comparable with aerosol scattering, since the scattering measurements are affected by the aerosol size distribution?

While these variables are not directly comparable, we find that the discussion by Fu et al. (2016) to be useful in understanding the role of aerosols in $Hg^{II}$ formation in the particulate form. However, we appreciate that the conclusions proposed by Fu et al. (2016) may not mean the same thing in our study because of the differences in how aerosols were quantified, and therefore we added to the text, "However, it is difficult to directly and quantitatively compare aerosol number concentrations reported by Fu et al. (2016) with the aerosol scattering measurements made at SPL, given that scattering affected by aerosol size distribution," (L726).

Technical Comments:

-L151- should this be 1.31± 0.9 or 0.09 ng m-3? Check Table 1 as well.

The values have been corrected in this line and in Table 1.

-L190 - I don't understand what the last part of the sentence is adding (There also are 6 months of data in 2022)

We simplified this sentence, which now states "Analyses in the present manuscript focus specifically on measurements made from March 13, 2021 to September 15, 2021 and from March 3, 2022 to September 15, 2022 to encompass two complete six-month periods," (L221).

-L294 - what does data were excluded listwise mean?

We edited this sentence to say: "Data were excluded listwise by the model, meaning all data for a given hourly timestamp were removed from the model if one or more variables had no data. This technique ensured that there were no missing data in the input dataset, but reduced the total number of timestamps in included in each input dataset by 40% in spring 2021 (n = 1169), 55% in summer

2021 (n = 1147), 29% in spring 2022 (n = 1575), 33% in summer 2022 (n = 1729), and 31% in spring–summer 2022 (n = 3299)," (L328).

-L309 - statistically significant differences for which means?

We added clarification here: "Overall, mean $Hg^0$ concentrations varied minimally from 2021 (1.27 ± 0.13 ng m$^{-3}$) to 2022 (1.25 ± 0.11 ng m$^{-3}$), from spring to summer in each year, or from one season to that same season in the following year, even though t-tests for comparisons of seasonal means all indicated statistically significant differences (p < 0.01)," (L343).

-L319 - Reference Table 1 for this section?

A reference to Table 1 was added here.

-L637-640 - can this sentence be clarified? As it was just mentioned that local precipitation can play a role

This sentence was clarified to say, "The higher RH during non-event times concurrent with lower $Hg^{II}$ concentrations could indicate greater wet deposition or cloud droplet scavenging of $Hg^{II}$ than during event periods," (L758).

**Reviewer 2:**

The study by Derry et al. applied a new speciated atmospheric Hg measurement technique to measure atmospheric Hg0 and Hg(II) at 10 min interval at SPL in spring and summer 2021 and 2022. The authors combined the speciated atmospheric Hg observations with ancillary parameters, to investigate the concentrations and origins of atmospheric Hg(II) in the free troposphere. I overall agree with the measurement technique and interpretation of the observational data. The manuscript is also organized in a good manner. Overall, I feel that this study is good at concrete observations but appears to be lack of providing novel findings as compared with previous studies, and this should be improved. I have several moderate concerns which hope be to considered before the publication in ACP.

Line 27: the temporal behavior similar to previous work is not clear to me. Please specified the temporal behavior.

We reworded this sentence to be more specific. It now reads, "Oxidized Hg concentrations displayed similar diel and episodic behavior to previous work at SPL, but were approximately three times higher in magnitude due to improved measurement accuracy," (L27).

Line 32-33: In addition to the deposition during transport, the slope higher than -1.0 might be also caused by the lower atmospheric THg at high-altitude and removal of Hg(II) by cloud droplets,….

We considered the vertical profile of Hg, where THg decreases with altitude while the relative amount of $Hg^{II}$ increases, which we discuss in further detail in Section 3.2.1. Based on the results of the HYSPLIT-STILT model which show air masses associated with high $Hg^{II}$ as originating in the low to mid-free troposphere, we felt that it was more likely that the lack of mass closure was caused by loss during transport, as there is little evidence for UT/LS influence. It is possible that cloud droplets could contribute to $Hg^{II}$ removal, and we have therefore added this point to the abstract and to our discussion in Section 3.2. We also adjusted the abstract to clarify that our conclusion about free tropospheric oxidation and transport was based not only on the slope, but other lines of evidence from trace gas data and model results.

Line 118: please specify the intervals for changing the cation membranes.

No change was made, as we already stated that the cation-exchange membranes were replaced every two weeks (L155).

Line 121-123: I suggest the author mention the other factors the might contribute the analytical uncertainty of the method, e.g., would cloud water Hg(II) be collected by the thermal converter under cloudy or rainy conditions? A 2.5 min analytical interval for Hg detection using the Tekran 2537 analyzer would generate large analytical uncertainty in Hg detection due to low Hg loading?

A full discussion and analysis of the uncertainty of the dual-channel Hg measurement system is available in Elgiar et al. (2024), however, we included additional information about that factors that went into the uncertainty analysis: "The percent standard uncertainty for $Hg^0$ and $Hg^{II}$ with the dual-channel system, which takes into account the uncertainty budget for the Tekran 2537X analyzer (following the methodology of Brown et al. (2008)), and also for the performance of the dual-channel component of the system, was 8% (Elgiar et al., 2024)," (L163).

Line 142-156: a discussion on the negative values based on the method is excellent. The strategy that excluding these negative values, however, is not convinced to me. This has a potential to reduce the reported mean concentrations during the whole study period, although the authors claim that this did not change the statistics of the data. I suppose the negative values should be mainly related to the low Hg(II) during these periods. I therefore suggest to define these negative values to be 0.

The negative values in the data are likely due to error in the instrumentation given the abrupt nature in which they occurred. They are very likely not real zero concentration measurements given that no other trace gas or aerosol measurement changed simultaneously during this period, and therefore we chose to exclude them. Replacing them with zeros would artificially bias the data toward values that we do not know to be real. As the negative values comprised ~2% of the total Hg data, excluding them did not meaningfully impact the seasonal mean Hg concentrations as already stated in the manuscript.

Table 1: it is better to show that negative values were excluded in the caption.

The exclusion of negative values has been added to the Table 1 caption.

Section 2.3.3: the intervals that calculating a new trajectory should be added.

No change was made, as Section 2.3.3 already states that trajectories were calculated at 3-hour intervals (L280).

Line 306-312: the description of the temporal variations is opposite to the statement in the abstract.

We adjusted the statement in the abstract to add clarity, which now states, "Oxidized Hg concentrations displayed similar diel and episodic behavior to previous work at SPL, but were approximately three times higher in magnitude due to improved measurement accuracy," (L27).

Section 3.1.2.: in addition to the power plant emissions, would other anthropogenic sources affect the atmospheric Hg? A simple to detailed discussion might be of interests.

Previous work at SPL has demonstrated that the site receives relatively little anthropogenic pollution due to its high elevation and remote location (Hallar et al., 2016; Obrist et al., 2008), aside from the three upwind power plants. These measurements were also cross-confirmed using the U.S. Environmental Protection Agency's National Emissions Inventory, which showed very few additional point sources of mercury or sulfur dioxide upwind of SPL. It is therefore unlikely that other anthropogenic sources would affect the atmospheric Hg measured at SPL, particularly considering the vertical transport profile of the air masses containing elevated $Hg^{II}$.

Line 360-367: a comparison between this study and previous observations to show hemispheric Hg decline has many uncertainties, considering that these sites were impacted by different regional sources or investigated by different times. In addition, all the references supporting their hypothesis did not report decline over the past 15 years. Other references regarding the hemispheric GEM trends should be added.

We edited this paragraph to further discuss the uncertainties in analyses of long-term atmospheric Hg trends, relying on recent reviews of the literature that discuss the discrepancies in these trends and their possible causes, such as Lyman et al. (2020a) and Sonke et al. (2023), and we added further long-term Hg data results from Weigelt et al. (2015).

416-431: the diurnal distributions of Hg(II) is in contrast with most high-altitude observations. Currently, the interpretation is not very sufficient. I would suggest the authors to add other proxies (e.g., O3, humidity, CO, SO2) in Figure 2, which would be helpful for better understand the controls. The author mainly focused on the air mass origins, and would local Hg(II)

production or removals contribute the diurnal trend in Hg(II). A local wind system together with large-scale wind field would help to diagnose the potential effects?

We added a paragraph to further discuss the potential causes of the diel pattern of $Hg^{II}$ at SPL, including more analyses we performed (L476). In addition, we examined the local wind data from SPL to determine whether wind speed or direction could have been contributing to this pattern, but no distinct patterns were seen aside from the generally prevailing westerly winds that have already been shown in previous work by Hallar et al. (2016). Additionally, no notable diel patterns were seen in the day vs night analyses for other trace gases and aerosols. We have added box plots demonstrating this to the Appendix for ozone and $PM_1$ aerosol scattering. However, given the available data and information in this study, we are unable to conclusively determine the cause of this pattern.

Line 541-542: where did these oxidations occur? In the continental or oceanic free troposphere?

Given the current understanding of the reaction rate of atmospheric mercury oxidation (Shah et al., 2021; Castro et al., 2022; Lee et al., 2024) and the results of the HYSPLIT-STILT model, it is likely that oxidation is occurring far enough upwind to be in the oceanic free troposphere. However, determining specifically where in transport the oxidation occurred is beyond the scope of this study.

Line 558: the study by Fu et al. did not attribute many Hg(II) enrichment event to air masses from UT/LS, instead, to the air masses from lower and middle FT over the North Atlantic Ocean.

The study by Fu et al. (2016) attributed many of the events of enhanced $Hg^{II}$ to the low to mid-FT over the North Atlantic, however the events that were attributed to the UT/LS are the ones being directly compared to the high $O_3$ events in this study. In this instance, we are referring specifically to the events where the authors reported co-enhancements of $Hg^{II}$ with $O_3$, and simultaneous decreases of pollution tracers.

Line 570-583: the explanation of the higher O3 concentrations during Hg(II) events is confuse to me. if these high O3 were related to PBL pollutions and biomass burning, would these sources also contribute to high Hg(II) events? This contradicts the major conclusion of this study that Hg(II) is mainly produced by atmospheric processes. The vertical profile in O3 might be evident in the continental atmosphere, even below upper free troposphere. The authors should also consider that O3 and Hg(II) have a similar production mechanism (e.g., photochemical process under dry air conditions) or removal processes.

We clarified our discussion of concurrent enhancements of ozone and $Hg^{II}$ to say, "Alternatively, $O_3$ could have been picked up from the North American PBL and mixed with free tropospheric air as the air masses traveled over the continent before arriving at SPL, or $O_3$ could have been produced in conjunction with $Hg^{II}$, as their chemical production mechanisms both involve

photochemical processes in dry air conditions," (L667). As discussed in the manuscript, biomass burning pollution is one possible source of elevated ozone at SPL, however, it is unlikely that ozone produced from the products of a biomass burning plume would contribute directly to the production of the elevated $Hg^{II}$ at SPL because of the rate of the oxidation mechanism, which operates on the scale of weeks (Shah et al., 2021; Castro et al., 2022). Therefore, local or regional ozone production related to biomass burning would likely not contribute directly to $Hg^{II}$ production by the time it reached SPL, and elevated $O_3$ may be coincidental to the elevated $Hg^{II}$. We added an additional point about this, stating that, "it is also possible that co-emitted halogens or aerosols in smoke plumes led to faster oxidation or scavenging of $Hg^0$ than in clean air, leading to elevated $Hg^{II}$ and PBM. Further work is needed to characterize the concentrations and chemistry of $Hg^0$ and $Hg^{II}$ in smoke plumes using verified measurement methods," (L683).

New References:

Brown, R. J., Brown, A. S., Yardley, R. E., Corns, W. T., Stockwell, P. B.: A practical uncertainty budget for ambient mercury vapour measurement, Atmospheric Environment, 42, 2504–2517, https://doi.org/10.1016/j.atmosenv.2007.12.012, 2008.

Friedli, H. R., Radke, L. F., Lu, J. Y., Banic, C. M., Leaitch, W. R., and MacPherson, J. I.: Mercury emissions from burning of biomass from temperate North American forests: laboratory and airborne measurements, Atmospheric Environment, 37, 253–267, https://doi.org/10.1016/S1352-2310(02)00819-1, 2003.

Fu, X., Jiskra, M., Yang, X., Marusczak, N., Enrico, M., Chmeleff, J., Heimbürger-Boavida, L.-E., Gheusi, F., and Sonke. J. E.: Mass-Independent Fractionation of Even and Odd Mercury Isotopes during Atmospheric Mercury Redox Reactions, Environ. Sci. Technol., 55, 10164–10174, https://doi.org/10.1021/acs.est.1c02568, 2021.

Obrist, D., Moosmüller, H., Schürmann. R., Antony Chen, L.-W., and Kreidenweis, S. M.: Particulate-Phase and Gaseous Elemental Mercury Emissions During Biomass Combustion: Controlling Factors and Correlation with Particulate Matter Emissions, Environ. Sci. Technol., 42, 3, 721−727, https://doi.org/10.1021/es071279n, 2007.

Saiz-Lopez, A., Acuña, A. U., Mahajan, A. S., Dávalos, J. Z., Feng, W., Roca-Sanjuán, D., Carmona-García, J., Cuevas, C. A., Kinnison, D. E., Gomez Martín, J. C., Francisco, J. S., and Plane, J. M. C.: The Chemistry of Mercury in the Stratosphere, Geophysical Research Letters, 49, https://doi. org/10.1029/2022GL097953, 2022.

Sonke, J. E., Angot, H., Zhang, Y., Poulain, A., Björn, E., and Schartup, A.: Global change effects on biogeochemical mercury cycling, Ambio, 52, 853–876, https://doi.org/10.1007/s13280-023-01855-y, 2023.

Weigelt, A., Ebinghaus, R., Manning, A. J., Derwent, R. G., Simmonds, P. G., Spain, T. G., Jennings, S. G., and Slemr, F.: Analysis and interpretation of 18 years of mercury observations since 1996 at Mace Head, Ireland, Atmospheric Environment, 100, 85–93, https://doi.org/10.1016/j.atmosenv.2014.10.050, 2015.

Deleted References:
Weiss-Penzias et al. "The effect of precipitation washout on oxidized mercury concentration as quantified by two types of atmospheric mercury measurement systems." This manuscript is still in preparation and will not available for citation at the time of re-submission